# mRNA decapping is an evolutionarily conserved modulator of neuroendocrine signaling that controls development and ageing

Fivos Borbolis[1,2], John Rallis[1,2], George Kanatouris[1,2], Nikolitsa Kokla[1,2], Antonis Karamalegkos[1,2], Christina Vasileiou[1,3], Katerina M Vakaloglou[1], George Diallinas[2], Dimitrios J Stravopodis[2], Christos G Zervas[1]*, Popi Syntichaki[1]*

[1]Biomedical Research Foundation of the Academy of Athens, Center of Basic Research, Athens, Greece; [2]Department of Biology, School of Science, National and Kapodistrian University of Athens, Athens, Greece; [3]Department of Molecular Biology and Genetics, Democritus University of Thrace, Alex/polis, Greece

**Abstract** Eukaryotic 5′−3′ mRNA decay plays important roles during development and in response to stress, regulating gene expression post-transcriptionally. In *Caenorhabditis elegans*, deficiency of DCAP-1/DCP1, the essential co-factor of the major cytoplasmic mRNA decapping enzyme, impacts normal development, stress survival and ageing. Here, we show that overexpression of *dcap-1* in neurons of worms is sufficient to increase lifespan through the function of the insulin/IGF-like signaling and its effector DAF-16/FOXO transcription factor. Neuronal DCAP-1 affects basal levels of INS-7, an ageing-related insulin-like peptide, which acts in the intestine to determine lifespan. Short-lived *dcap-1* mutants exhibit a neurosecretion-dependent upregulation of intestinal *ins-7* transcription, and diminished nuclear localization of DAF-16/FOXO. Moreover, neuronal overexpression of DCP1 in *Drosophila melanogaster* confers longevity in adults, while neuronal DCP1 deficiency shortens lifespan and affects wing morphogenesis, cell non-autonomously. Our genetic analysis in two model-organisms suggests a critical and conserved function of DCAP-1/DCP1 in developmental events and lifespan modulation.

*For correspondence: czervas@bioacademy.gr (CGZ); synticha@bioacademy.gr (PS)

**Competing interests:** The authors declare that no competing interests exist.

## Introduction

Addition of an $m^7G$ cap structure at the 5′-end of nascent transcripts (capping) is a universal modification of eukaryotic mRNAs, which takes place co-transcriptionally, briefly after incorporation of the first nucleotides (*Moteki and Price, 2002*). $m^7GpppN$ (where N is the first transcribed nucleotide) cap is a major *cis* regulatory element that interacts with multiple binding partners to promote pre-mRNA processing, to guide nuclear export and eventually to drive the formation of the translation apparatus in the cytoplasm. Removal of the 5′-cap structure (decapping), predominately occuring after shortening of the poly(A) tail (deadenylation), constitutes a pivotal event in mRNA lifecycle, since it pulls the transcript out of the translational pool and renders it susceptible to degradation by the highly conserved 5′−3′ exoribonuclease XRN1 (*Jones et al., 2012*). This cytoplasmic 5′−3′ mRNA decay pathway can be also triggered in a transcript-specific manner through the interaction of mRNAs with miRNAs or specific RNA-binding proteins that regulate gene expression in response to various endogenous or external signals (*Borbolis and Syntichaki, 2015*).

Although for years, the common consensus stated that decapping is an irreversible process that commits transcripts to degradation, this view was overturned mainly by the discovery of mRNA

capping enzymes in the cytoplasm of diverse eukaryotic cells and the identification of capped mRNA ends that did not correspond to transcription start sites (*Trotman and Schoenberg, 2019*; *Affymetrix ENCODE Transcriptome Project and Cold Spring Harbor Laboratory ENCODE Transcriptome Project, 2009*). These findings introduced the concept of 'cap homeostasis' in gene expression, where cyclic decapping and recapping regulates translational activity of at least a subset of mRNAs (*Mukherjee et al., 2012*). In this setting, uncapped transcripts are stored in a translationally inactive, but stable form in cytoplasmic mRNP granules termed P-bodies, where they coincide with decapping factors, translational repressors and components of the 5′−3′ mRNA decay and RNA interference machineries (*Mukherjee et al., 2012*; *Standart and Weil, 2018*). Despite their composition though, P-bodies seem to be primarily involved in the coordinated storage of mRNA regulons rather than mRNA decay (*Hubstenberger et al., 2017*; *Standart and Weil, 2018*; *Luo et al., 2018*). Such discoveries underscored the role of mRNA decapping as a potent post-transcriptional regulator of gene expression. Interestingly, most decapping defects have been linked to severe neurological disorders, highlighting the importance of decapping activity in neuronal cells, where spatiotemporal control of translation is of supreme importance and the presence of P-bodies and related mRNP granules is more frequent (*Jiao et al., 2006*; *Ahmed et al., 2015*; *Donlin-Asp et al., 2017*).

To exert the appropriate control over mRNA decay, eukaryotes have developed a multi-protein decapping complex, formed around the catalytic subunit DCP2 and its essential co-factor DCP1. Together DCP1/DCP2 form a poorly active holoenzyme that needs to interact with more auxiliary factors in order to work efficiently and to provide substrate specificity (*Charenton and Graille, 2018*). Accumulating evidence suggests that both availability and activity of DCP1 play a major regulatory role in cap homeostasis and mRNA decay, controlling the balance between decapping, translation and degradation. FRAP experiments in human cells have shown that the fraction of P-body localized DCP2 is rather fixed but DCP1 molecules can be rapidly exchanged between P-bodies and the cytoplasm (*Aizer et al., 2008*), while P-body formation is induced by DCP1 overexpression, endogenous or environmental stressors and ageing (*Kedersha et al., 2005*; *Rousakis et al., 2014*).

We have previously shown that depletion of *dcap-1* or *dcap-2*, the *C. elegans* orthologs of human DCP1 and DCP2 respectively, can affect various aspects of worm physiology, provoking reduced fertility, developmental defects, increased sensitivity to stress and shortened lifespan (*Rousakis et al., 2014*). Particularly, neuronal DCAP-1 has been found to influence critical developmental events that modulate the worm's growth rate under optimal conditions, and affect its ability to form stress resilient dauer larvae at high temperature, partially by affecting insulin/IGF-like signaling (IIS) (*Borbolis et al., 2017*). IIS is an evolutionarily conserved pathway that participates in various cellular processes and regulates lifespan across species, from invertebrates to mammals. *C. elegans*' genome encodes for ~40 insulin-like peptides (ILPs) that initiate IIS by binding to insulin/IGF-like receptor (DAF-2) and regulate the localization/function of the downstream target DAF-16/FOXO transcription factor (*Sun et al., 2017*). Previous work has shown that IIS mediates the communication of multiple tissues to regulate the rate of ageing; while it is DAF-2 activity predominantly in the nervous system that regulates lifespan, DAF-16 acts mainly in the intestine to confer DAF-2-regulated longevity (*Wolkow et al., 2000*; *Libina et al., 2003*). This kind of inter-tissue communication is considered a vital part of organismal ageing in *C. elegans*, *Drosophila* and mammals. Especially the bidirectional communication between nervous system and intestinal tissue, known as the brain-gut axis, has recently emerged as an important regulatory circuit that controls lifespan across species (*Zhang et al., 2018*; *Westfall et al., 2018*).

In the current study, we provide evidence that DCAP-1 activity in the nervous system of *C. elegans* participates in the regulation of a neurosecreted ILP, which ultimately affects the function of DAF-16 transcription factor in distal tissues. In consistence with the central role of IIS in ageing and stress response, we show that neuronal overexpression of DCAP-1 increases lifespan and enhances stress resistance of aged worms. Moreover, we unveil evolutionarily conserved functions of neuronal DCP1 activity in the fruit fly *Drosophila melanogaster*. Our observations suggest a critical role for neuronal mRNA decapping during normal ageing and provide insights into the underlying mechanism.

## Results

### Neuronal decapping regulates lifespan

We have previously reported that expression of a *dcap-1::gfp* transgene driven by the native *dcap-1* promoter, can fully rescue all the defects of *dcap-1(tm3163)* mutants, including their short lifespan (*Rousakis et al., 2014*). In order to explore the tissue-specific requirement of DCAP-1 in lifespan regulation, we generated a series of transgenic lines that overexpress this *dcap-1::gfp* fusion gene in different tissues of *dcap-1* worms. We found that restoring the function of *dcap-1* gene specifically in the nervous system (*unc-119p-synEx293 array*) could significantly increase lifespan of *dcap-1* mutants, rendering it comparable to that of wild type (wt/N2) animals at 20°C (*Figure 1A*). Similar results were obtained by creating a second transgenic strain that expresses the fusion under the control of a different pan-neuronal promoter (*rab-3p-synEx345*) (*Figure 1—source data 1*). This effect was also evident at 25°C (*Figure 1E*), where *dcap-1(tm3163)* phenotypes are exaggerated and lifespan is even shorter (*Rousakis et al., 2014*). In contrast, overexpression of *dcap-1::gfp* in the epidermis (*col-10p-synEx364*) or in muscles (*hlh-1p-synEx326*) only modestly improved lifespan at 20°C, and had practically no effect at 25°C (*Figure 1B,C,F,G*). An intermediate result was observed in *dcap-1* mutants overexpressing *dcap-1::gfp* in the intestine (*ges-1p-synEx328*) at both 20°C and 25°C (*Figure 1D,H*).

The results of our complementation analysis suggest that *dcap-1* gene activity affects lifespan, albeit to a different extent depending on the tissue. The fact that overexpression of this *dcap-1::gfp* transgene exclusively in the nervous system was able to fully rescue the short-lived mutant phenotype, suggests a strong positive regulation of lifespan even though *dcap-1* is missing from other tissues. We therefore examined the impact of overexpressing all tissue-specific *dcap-1::gfp* fusions on the lifespan of otherwise wt animals. As shown in *Figure 2A*, neuronal overexpression of *dcap-1::gfp* under the *unc-119* promoter (*synEx293*) resulted in a significant increase of both median and maximum lifespan of N2 worms, by ~16% and~26%, respectively (*Figure 2—source data 1*). Comparable results were obtained when expression was driven by *rab-3* promoter (*synEx345*,~13% and~18%

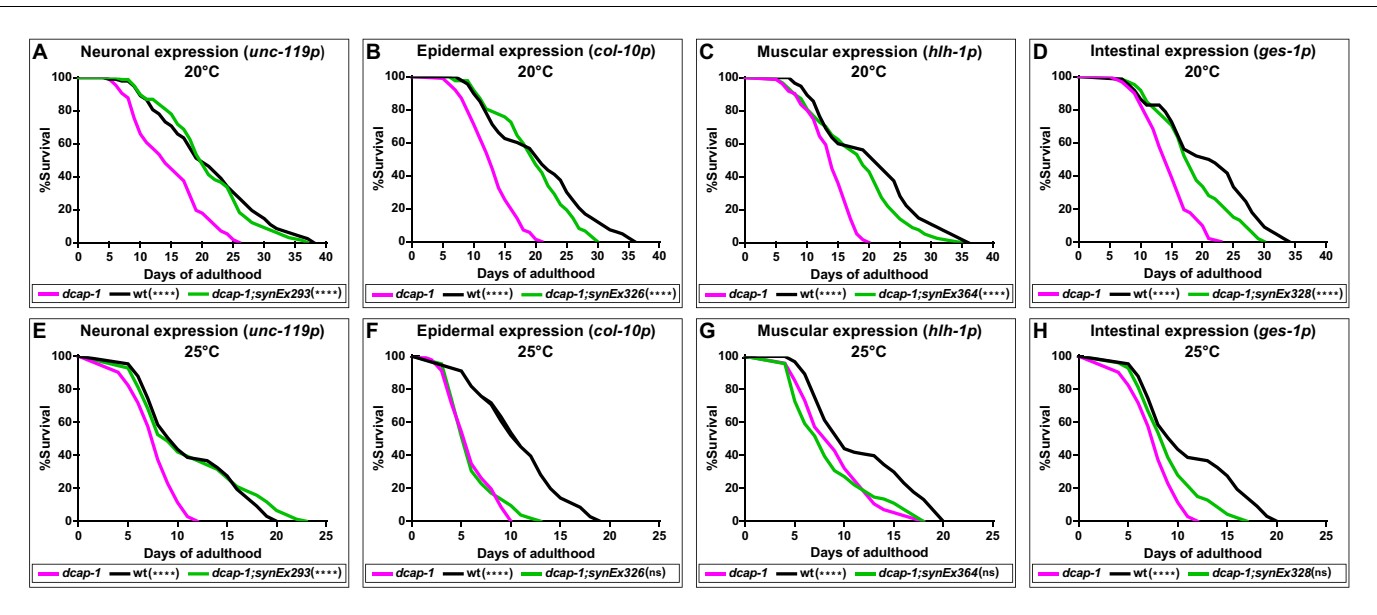

**Figure 1.** Neuronal restoration of *dcap-1* rescues the short lifespan of *dcap-1* mutants. Lifespan of *dcap-1* worms overexpressing a *dcap-1::gfp* fusion in neurons, epidermis, muscles or intestine at (**A–D**) 20°C and (**E–H**) 25°C. Statistical significance is calculated in comparison to *dcap-1*. ****p<0.0001. Log-rank (Mantel-Cox) test.

The online version of this article includes the following source data for figure 1:

**Source data 1.** Lifespan replicates of *dcap-1(tm3163)* worms that overexpress a *dcap-1::gfp* fusion tissue-specifically at 20°C.

**Source data 2.** Lifespan replicates of *dcap-1(tm3163)* worms that overexpress a *dcap-1::gfp* fusion tissue-specifically at 25°C.

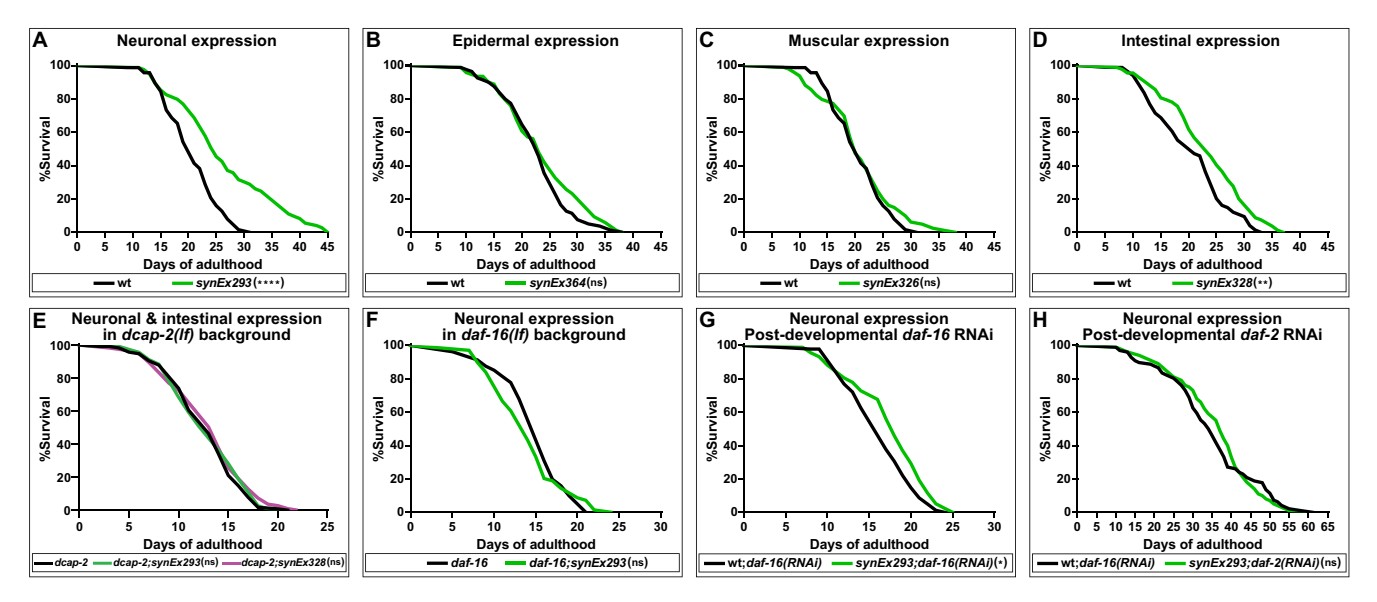

**Figure 2.** Neuronal overexpression of *dcap-1* extends lifespan through mRNA decapping, DAF-16/FOXO and IIS. Lifespan at 20°C of (**A–D**) wt worms overexpressing a *dcap-1::gfp* fusion in neurons, epidermis, muscles or intestine, (**E**) worms overexpressing neuronal or intestinal *dcap-1::gfp* in *dcap-2* mutant genetic background, (**F**) worms overexpressing neuronal *dcap-1::gfp* in *daf-16* mutant genetic background, (**G**) worms overexpressing neuronal *dcap-1::gfp* under post-developmental *daf-16* RNAi knockdown, (**H**) worms overexpressing neuronal *dcap-1::gfp* during post-developmental daf-2 RNAi knockdown. *p<0.05, **p<0.01, ****p<0.0001. Log-rank (Mantel-Cox) test.

The online version of this article includes the following source data for figure 2:

**Source data 1.** Lifespan replicates of worms that overexpress a *dcap-1::gfp* fusion tissue-specifically in wtgenetic background at 20°C.
**Source data 2.** Lifespan replicates of worms depleted of *daf-2* or *daf-16* at 20°C.
**Source data 3.** Lifespan replicates of *dcap-2(ok2023)* worms that overexpress a *dcap-1::gfp* fusion tissue-specifically at 20°C.

lifespan increase) (*Figure 2—source data 1*). In agreement with our observations from the complementation analysis, expression in either muscles (*synEx326*) or epidermis (*synEX364*) did not have any effect on the longevity of wt animals, whereas a smaller extension of lifespan (~8% in median and ~10% in maximum) was achieved by overexpressing *dcap-1* in the intestine (*synEx328*) (*Figure 2B, C, D*). Most importantly, lifespan extension in all cases was accompanied by an analogous delay in the onset of age-dependent movement impairment, which is considered a reliable marker of ageing (*Son et al., 2019*), with worms remaining responsive to touch longer than their respective controls (*Video 1*). Therefore, neuronal (and to a lesser extent intestinal) overexpression of *dcap-1* can extend the lifespan and healthspan of both *dcap-1* and wt animals. Moreover, the mechanism underlying their longevity completely depends on mRNA decapping, as the effect of neuronal or intestinal overexpression was absent in *dcap-2(ok2023)* mutants that lack decapping catalytic subunit activity (*Figure 2E*).

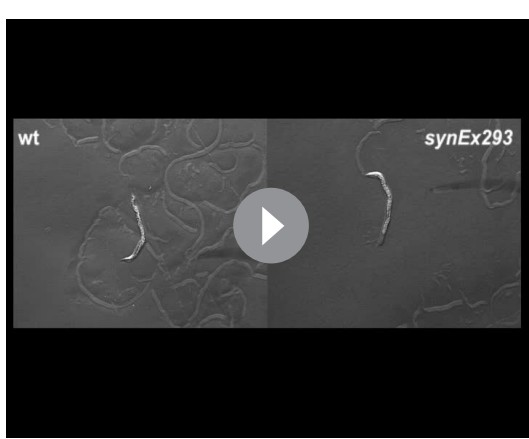

**Video 1.** Neuronal overexpression of *dcap-1* delays the onset of age-dependent movement impairment. Video of 20 days old worms.
https://elifesciences.org/articles/53757#video1

## Neuronal decapping regulates lifespan through IIS and DAF-16/FOXO

Control of lifespan by genetic manipulations in the nervous system often involves the action of neuroendocrine signals that adjust the function of longevity-regulating genetic pathways, most of which converge at DAF-16 transcription factor (*Sun et al., 2017*). To evaluate the effect of *daf-16* deficiency on the longevity of animals overexpressing neuronal *dcap-1*, we transferred *synEx293* (*unc-119p::dcap-1::gfp*) transgene in worms homozygous for the *daf-16(mu86)* null allele. Neuronal *dcap-1* overexpression under such DAF-16-deficient conditions did not have any impact on longevity, with *daf-16(mu86);synEx293* animals living as *daf-16(mu86)* mutants (*Figure 2F*). Even post-developmental knockdown of *daf-16* by RNAi was sufficient to largely suppress the long lifespan of worms overexpressing neuronal *dcap-1* (*Figure 2G*). The little residual longevity observed in these animals is most likely to reflect the inability of post-developmental RNAi to completely eliminate DAF-16 activity. Collectively, our results suggest that the effect of neuronal DCAP-1 in adult life is mediated by DAF-16. It is noteworthy that we did not find any significant difference in lifespan between double *dcap-1(tm3163);daf-16(mu86)* and single *daf-16(mu86)* mutants, suggesting that loss of *dcap-1* reduces lifespan through negative regulation of DAF-16 activity or that *daf-16* is epistatic to *dcap-1* in lifespan determination (*Figure 2—source data 2*).

Given that *C. elegans* neurons are fairly refractory to RNAi (*Kamath et al., 2003*) our data suggest that neuronal DCAP-1 acts in a cell non-autonomous manner, involving signals that influence DAF-16 behavior in distal tissues. The prevalent pathway that integrates intercellular signals to regulate DAF-16 activity, with a great impact on longevity, is IIS. We thus reassessed the lifespan of *synEx293* (*unc-119p::dcap-1::gfp*) animals, under conditions of reduced IIS, caused by post-developmental knockdown of insulin/IGF-1 receptor gene *daf-2*. As expected, *daf-2* RNAi resulted in extreme longevity in wt animals but completely masked the effect of neuronal *dcap-1* overexpression, which failed to further extend lifespan (*Figure 2H*). This lack of an additive effect indicates that decapping in the nervous system acts through IIS to regulate DAF-16 activity.

## Neuronal *dcap-1* regulates INS-7 to confer longevity

In *C. elegans*, DAF-2 receives input from at least 40 insulin-like peptides (ILPs) that act as agonists or antagonists to regulate stress resistance, reproduction, development and ageing (*Fernandes de Abreu et al., 2014*). Expression of ILPs has been detected in diverse tissues, but interestingly, those that regulate lifespan are expressed mainly in the nervous system and often act in non-neuronal cells to influence DAF-16 activity and ageing (*Artan et al., 2016*; *Murphy et al., 2007*; *Libina et al., 2003*). We therefore examined the effect of *dcap-1* depletion or neuronal *dcap-1* overexpression on *ins-7, daf-28, ins-33, ins-1* and *ins-6* ILPs with a well-established role in neuroendocrine lifespan control (*Artan et al., 2016*; *Murphy et al., 2007*). Our qRT-PCR analysis revealed that although *daf-28, ins-33, ins-1 and ins-6* mRNA levels are similar among all genetic backgrounds, *ins-7* abundance is almost doubled in young *dcap-1* animals (1 day old), and even more upregulated in mid-aged mutants (9 days old), compared to wt worms of the same age ( *Figure 3A*). Importantly, mRNA levels of *ins-7* were significantly lower in mid-aged *synEx293* worms (that overexpress neuronal *dcap-1*) paralleled to wt (*Figure 3A*). INS-7 is a DAF-2 agonist, expressed predominantly in neurons and scarcely in the intestine of young animals, but in aged worms intestinal expression is considerably enhanced (*Murphy et al., 2007*). In line with this, our results show an increase of *ins-7* mRNA abundance in aged worms of all genotypes. However, this age-dependent increment from days 1 to 9 is significantly enhanced in *dcap-1* (seven times) compared to wt (5.5 times), and suppressed in neuronal *dcap-1* overexpressing worms (2.5 times).

In order to clarify whether DCAP-1 affects *ins-7* mRNA abundance transcriptionally or post-transcriptionally, we monitored the fluorescence of wt and *dcap-1* animals carrying a transcriptional *ins-7p::gfp* reporter (*wwEx66* array) (*Ritter et al., 2013*), at various ages of their adult lifespan (days 1 to 15). We detected fluorescence mainly in head neurons and some intestinal cells, with advanced age resulting in increased intestinal fluorescence for both strains (*Figure 3B—figure supplement 1*). Even so, total fluorescence intensity of *dcap-1;wwEx66* worms was greater at all examined ages compared to single *wwEx66*, with the difference fluctuating between 1.6- and 2.1-fold (*Figure 3B*). Such an increase in promoter activity argues in favor of DCAP-1 regulating *ins-7* expression at the level of transcription. In particular, we observed that this effect is restricted to the intestine of *dcap-1;wwEx66* worms, as there was no discernible effect on neuronal cells of the head region or

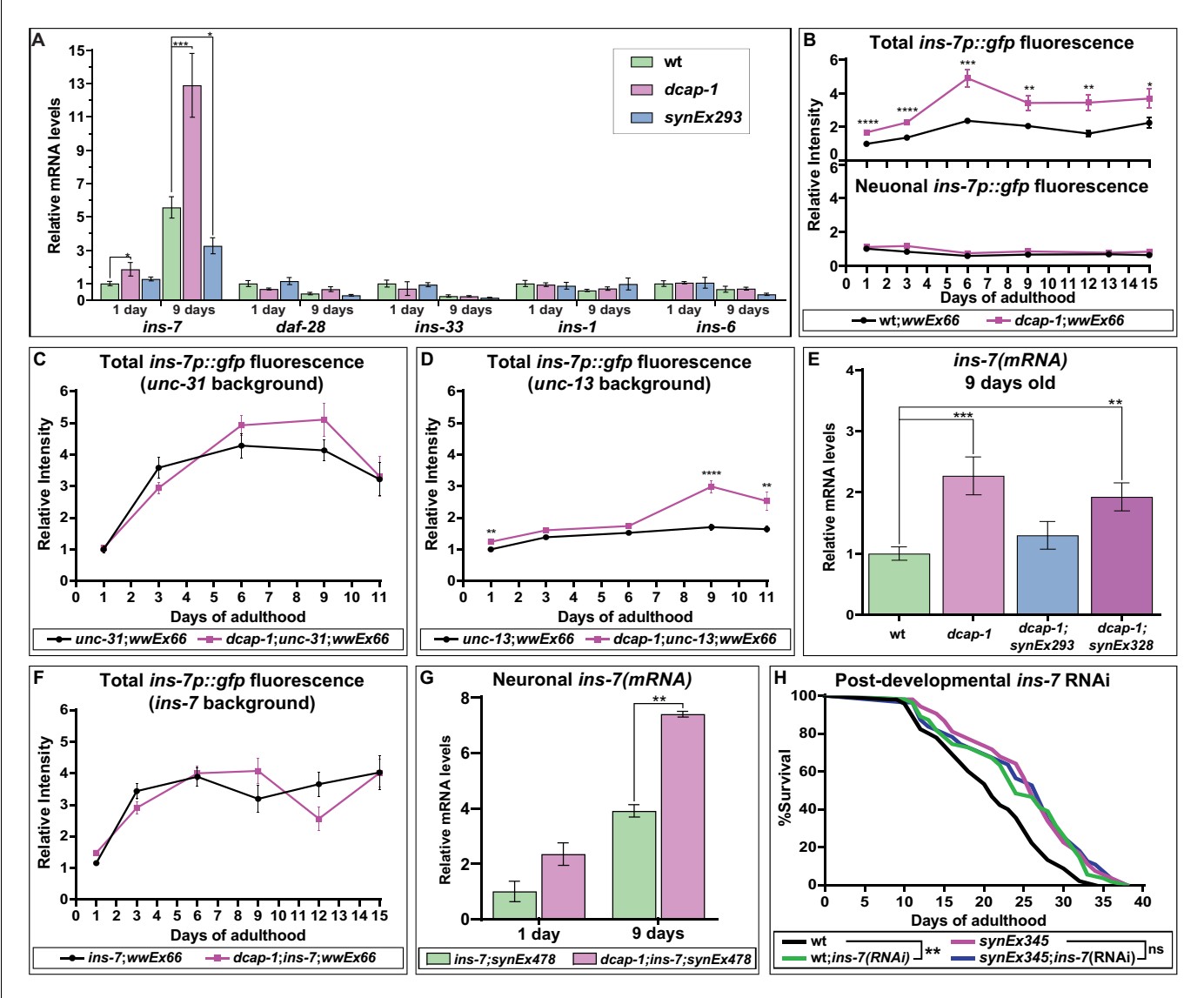

**Figure 3.** Neuronal *dcap-1* regulates *ins-7* expression to control longevity. (A) mRNA levels of lifespan-regulating ILPs in young (1 day old) and mid-aged (9 days old) wt, *dcap-1* and neuronal *dcap-1* overexpressing worms. (B) Fluorescence intensity of *ins-7p::gfp* reporter in whole worms or head neurons of wt and *dcap-1* animals. (C–D) Total fluorescence intensity of *ins-7p::gfp* reporter in neurosecretion defective *unc-31* or *unc-13* mutants. (E) *ins-7* mRNA levels in 9 days old worms overexpressing *dcap-1::gfp* in neurons (*synEx293*) or intestine (*synEx328*). (F) Total fluorescence intensity of *ins-7p::gfp* reporter in *ins-7* mutants. (G) *ins-7* mature mRNA/pre-mRNA ratio in *ins-7* and *dcap-1;ins-7* animals that carry a neuronally expressed *unc-119p::ins-7* transgene. (H) Lifespan of wt and neuronal *dcap-1* overexpressing worms sensitized for neuronal RNAi during post-developmental *ins-7* RNAi knockdown. *p<0.05, **p<0.01, ***p<0.001, ****p<0.0001. Error bars indicate mean ± SD. Unpaired t-test (A–G); Log-rank (Mantel-Cox) test (F). The online version of this article includes the following source data and figure supplement(s) for figure 3:

**Source data 1.** Lifespan replicates of worms exposed to *ins-7* RNAi at 20°C.
**Source data 2.** *Figure 3A, E and G* and *Figure 3—figure supplement 6*.
**Source data 3.** *Figure 3B, C, D and F* and *Figure 3—figure supplement 2A*.
**Figure supplement 1.** Transcriptional activity of *ins-7* promoter is significantly increased in *dcap-1* mutants.
**Figure supplement 2.** Mutation of *dcap-1* does not affect the expression of a *ges-1p::gfp* transgene.
**Figure supplement 3.** Transcriptional activity of *ins-7* promoter is not increased in *dcap-1* mutants when *unc-31* is missing.
**Figure supplement 4.** Transcriptional activity of *ins-7* promoter in *unc-31* mutants is further induced upon *daf-16* knockdown.
**Figure supplement 5.** Transcriptional activity of *ins-7* promoter is significantly increased in *dcap-1* mutants independently of *unc-13*.
**Figure supplement 6.** Transcriptional activity of *ins-7* promoter is not increased in *dcap-1* mutants when *ins-7* gene product is missing.
**Figure supplement 7.** The ratio of *eft-3* mature mRNA/pre-mRNA is not affected by *dcap-1* mutation.

elsewhere, in respect to *wwEx66* animals (*Figure 3B—figure supplement 1*). This induction is specific for *ins-7* promoter since intestinal fluorescence of a *ges-1p::gfp* transgene did not show an analogous increase in *dcap-1* background (*Figure 3—figure supplement 2*). Therefore, the observed differences in *ins-7* mRNA abundance between wt and *dcap-1* probably reflect transcriptional changes in the intestine.

Apart from acting as a DAF-2 agonist, *ins-7* is also negatively regulated by DAF-16 in intestinal cells, in sharp contrast to neurons where *ins-7* is not responsive to IIS (*Murphy et al., 2007*). Since IIS intertissue signaling is mainly initiated by neurons (*Libina et al., 2003*), we argued that induction of *ins-7p::gfp* expression in the intestine of *dcap-1* animals could be the response to an upsurge in neuronal release of ILPs. Indeed, we found that deletion of *unc-31*/CAPS (allele *e928*), which is required for the Ca$^{2+}$-mediated release of neuropeptide-carrying dense core vesicles (DCVs) (*Speese et al., 2007*), eliminates differences in *ins-7p::gfp* fluorescence intensity between *dcap-1 (tm3163);wwEx66* and single *wwEX66* worms at all examined ages (*Figure 3C—figure supplement 3*). This cannot be attributed to a ceiling effect in *ins-7* promoter activity, as *daf-16* RNAi further induced fluorescence in both backgrounds (*Figure 3—figure supplement 4*). In contrast, blocking the release of small clear vesicles (SCVs), and hence of neurotransmitters, by using *unc-13(e450)* mutant allele (*Madison et al., 2005*), generally limited the induction of *ins-7p::gfp* fluorescence with age in both genetic backgrounds (*Figure 3D vs* 3B), but was largely incapable to suppress the stimulatory effect of *dcap-1* depletion on *ins-7p* activity, especially in aged worms (*Figure 3D—figure supplement 5*). Accordingly, qRT-PCR analysis revealed that restoring *dcap-1* gene function exclusively in neurons (*dcap-1;synEx293*) was able to diminish *ins-7* mRNA abundance back to wt levels (*Figure 3E*), whereas restoration in the intestine (*dcap-1;synEx328*) only partially reduced *ins-7* mRNA levels (*Figure 3E*).

Altogether, our data show that DCAP-1 activity in neurons regulates *ins-7* expression in the intestine through a factor released by DCVs. Given its self-regulatory function, we considered that INS-7 could be this factor. Indeed, deletion *ok1573*, which abolishes *ins-7* expression, completely suppressed the effect of *dcap-1* depletion on *ins-7p::gfp* induction (*Figure 3F—figure supplement 6*). Since *ins-7* transcription was not altered in *dcap-1* neurons, we examined whether decapping could affect the stability of neuronal *ins-7* mRNA. Using an *unc-119p::ins-7* transgene in *ins-7* and *dcap-1; ins-7* mutants, we quantified the ratio of mature *ins-7* mRNA to its pre-mRNA, thus avoiding discrepancies due to chimeric expression of the transgene. As shown in *Figure 3G*, there was a two-fold induction of mature mRNA levels in both 1- and 9-day *dcap-1* adults compared to wt. This was not due to a general effect on mRNA maturation and stability, as it did not apply to the reference gene *eft-3* (*Figure 3—figure supplement 7*), indicating the selective stabilization of *ins-7* mRNA in neurons of decapping mutants.

Previous work has shown that INS-7 serves as a carrier of FOXO-to-FOXO intertissue signaling coordinating the rate of ageing (*Murphy et al., 2007*). Consequently, the deficit in *ins-7* induction during ageing in neuronal *dcap-1* overexpressing animals, could be the reason for their prolonged lifespan. To support this theory, we assessed the longevity of wt and *synEx345 (rab-3p::dcap-1::gfp)* worms under post-developmental *ins-7* RNAi that should eliminate differences in *ins-7* expression between strains. To achieve efficient systemic RNAi, both strains were genetically modified for enhanced neuronal RNAi (see Materials and methods). As expected, *ins-7* RNAi increased lifespan of wt, but failed to cause an additive effect and further promote longevity in *synEx345* animals (*Figure 3H*). These data suggest that low levels of INS-7 may trigger lifespan extension in worms overexpressing neuronal *dcap-1*.

## Neuronal decapping affects stress resistance in aged animals

It is well documented that stress resistance and longevity are frequently linked aspects of organismal physiology, with IIS and DAF-16 providing a link between them. Thus, we examined whether *dcap-1* depletion affects the translocation of DAF-16 to the nucleus during heat-shock by exposing wt and *dcap-1* young adults that carry a *daf-16::gfp* translational fusion (*muIs71*) at high temperature (35°C). Notably, although *muIs71* control animals started to show an increase in DAF-16::GFP positive nuclei after just 15 min of heat-shock (data not shown), no such response was evident in *dcap-1;muIs71* worms, unless we exposed them for more than 30 min, and the number of DAF-16::GFP nuclei per animal was significantly lower in *dcap-1* mutants compared to wt at all examined time points, including unstressed worms (*Figure 4A—figure supplement 1*). Therefore, *dcap-1* depletion seems to

hinder the translocation of DAF-16 to the nucleus, under normal conditions and the early stages of heat-shock; longer exposure times (over 60 min) ultimately surmounted this inhibitory effect, leading to almost complete translocation of DAF-16::GFP to the nucleus in both strains (data not shown). Conversely, neuronal overexpression of DCAP-1 (*synEx293*) in worms carrying a *daf-16a::rfp* translational fusion (*lpIs12*) revealed a significant increase in the number of red fluorescent nuclei per *synEx293;lpIs12* animal, compared to single *lpIs12* worms, even in the absence of stress (*Figure 4B—figure supplement 2*). The difference in DAF-16 nuclear accumulation between the two strains remained significant after 5 min heat-shock but started to diminish at 15 min, as nuclear enrichment is enhanced in both backgrounds (*Figure 4B—figure supplement 2*). The different fluorophores combined with variations in transgene copy-number could account for the differential kinetics of DAF-16a::GFP and DAF-16a::RFP translocation in the same background (*Figure 4— source data 1*).

Given that *dcap-1* depletion affects DAF-16 localization, we postulated that restoring *dcap-1* function in neurons could also rescue the reported mutant's sensitivity to stress (*Rousakis et al., 2014*). Unexpectedly, we found that neuronal rescuing of *dcap-1* by either pan-neuronal promoter (*synEx293/synEx345*) failed to alleviate the sensitivity of 1 day old *dcap-1* adults in either heat-sock or oxidative stress (*Figure 4C*). Accordingly, neuronal expression of *dcap-1* (*synEx293*) in wt background did not increase tolerance to heat shock or oxidative stress of young adult worms (*Figure 4D*). However, the ability of an organism to cope with stress declines with age and mechanisms that preserve cellular homeostasis and genome integrity are of great importance during ageing. In line with this notion, prolonged exposure to low concentration of sodium arsenite revealed that *synEx293* provides resistance to oxidative stress in ageing animals, which becomes evident after the 3rd day of adulthood, and is maintained throughout their lifespan (*Figure 4E*). Similarly, *synEx293* animals exhibited increased resistance when subjected to heat shock at the 9th day of their adult life, compared to their wt counterparts (*Figure 4F*). This implies that neuronal *dcap-1* overexpression provides increased stress resistance in old ages, probably as a result of a slower ageing rate.

## Expression levels of DCP1 affect lifespan in *D. melanogaster*

Given the strict evolutionary conservation of both the 5'−3' mRNA decay machinery and the neuronal mechanisms of longevity, we reasoned that the connection between neuronal decapping and ageing might not be restricted to *C. elegans*. We therefore studied this relationship in the fly *D. melanogaster*. Because null mutants of DCP1 (the ortholog of *dcap-1*) are pupal lethal (*Lin et al., 2006*), we specifically induced knock-down of DCP1 in the nervous system, by crossing two independent UAS:IRDCP1 lines with elav:GAL4, which is expressed in neurons at all developmental stages (*Robinow and White, 1988*). The subsequent neuron-specific knockdown of DCP1, verified by qRT-PCR (*Figure 5A*), resulted in a significant reduction of lifespan, as elav:GAL4 >UAS:IRDCP1 progeny of either UAS:IRDCP1 strain were extremely short-lived compared to their parental elav:GAL4 flies at 25°C, and at 29°C where UAS/GAL4 system is most effective (*Figure 5B, C*). While this is a strong hint that neuronal mRNA decapping is involved in lifespan regulation in *Drosophila*, reduced longevity of these animals could also arise from a general sickness due to malfunction of the decapping complex.

Next, we examined whether neuronal overexpression of DCP1 has the opposite effect and induces longevity, similarly to worms. We generated a UAS:DCP1:eGFP strain and crossed it with elav:GAL4 flies to achieve overexpression of DCP1:eGFP transgene in the nervous system. In contrast to our expectations, elav:GAL4 >UAS:DCP1:eGFP flies grown at 25°C or 29°C did not live longer than their parental elav:GAL4 strain (*Figure 5—source data 1*). However, given the implication of neuronal DCP1 in development, we argued that abnormally high expression caused by the transgene might be harmful during early life stages, masking or counteracting any potential lifespan-extending effect of DCP1 overexpression during adulthood. For this reason, we used GAL80[ts], a temperature-sensitive inducible GAL4 inhibitor, to switch on neuronal DCP1:eGFP transgene overexpression after completion of development. elav:GAL4 >UAS:DCP1:eGFP;GAL80[ts] flies were grown at the permissive temperature for GAL80[ts] (18°C) and transferred to the restrictive temperature (29°C) upon the onset of adulthood. Interestingly, these flies that overexpress neuronal DCP1 post-developmentally exhibited a significant increase of both median (~42%) and maximum (~27%) lifespan compared to their parental elav:GAL4 strain (*Figure 5D*). Thus, high levels of DCP1 in neurons (or neuron

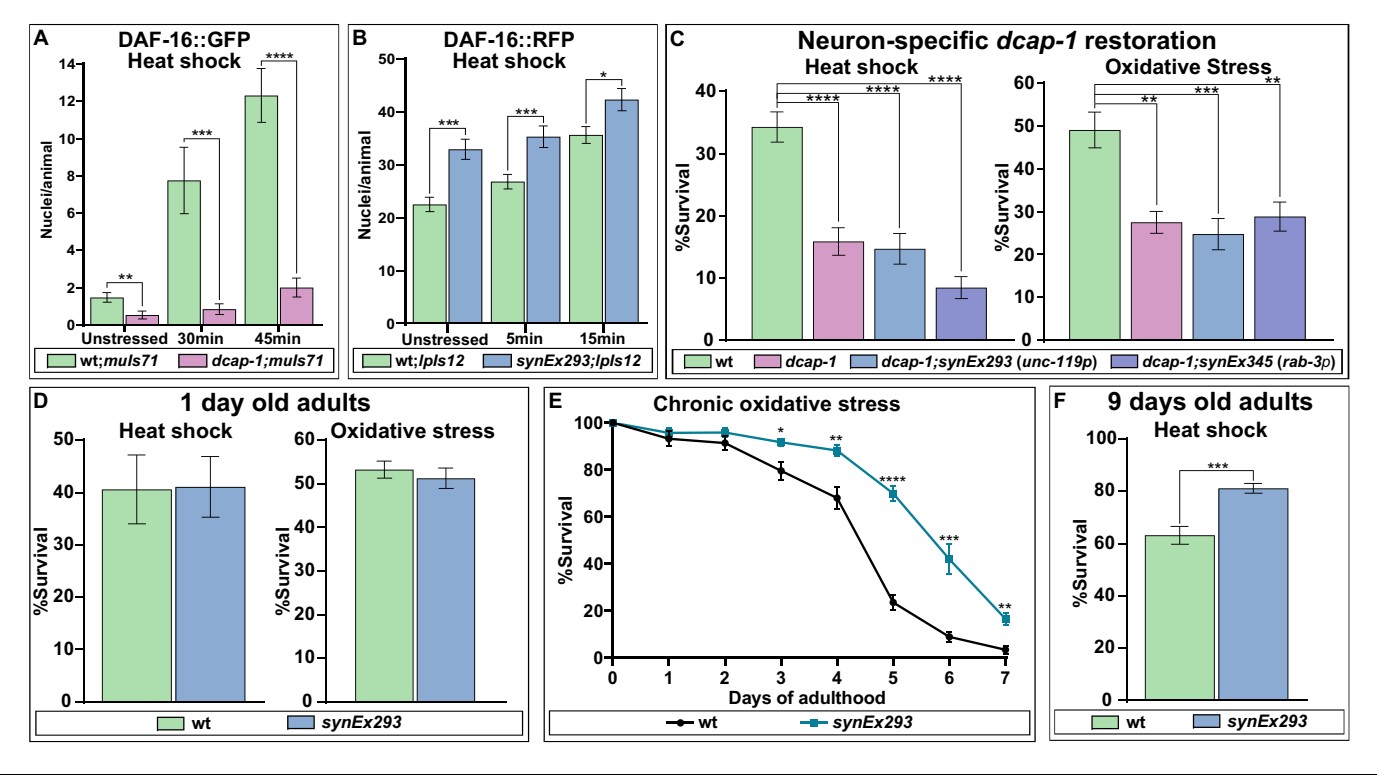

**Figure 4.** DCAP-1 affects DAF-16 localization and stress resistance. (**A**) Quantification of DAF-16a::GFP positive nuclei during early stages of heat shock response in 1-day-old wt and *dcap-1* mutant worms. (**B**) Quantification of DAF-16a::RFP positive nuclei during early stages of heat shock response in 1-day-old wt and neuronal *dcap-1::gfp* overexpressing worms. (**C**) Survival of 1-day-old *dcap-1* worms that express *dcap-1::gfp* in their neurons, after acute exposure to high temperature or high concentration of sodium arsenite. (**D**) Survival of 1-day-old wt worms overexpressing neuronal *dcap-1::gfp* after acute exposure to high temperature or high concentration of sodium arsenite. (**E**) Survival of wt worms overexpressing neuronal *dcap-1::gfp* during sustained exposure to low concentration of sodium arsenite. (**F**) Survival of 9 days old wt worms overexpressing neuronal *dcap-1::gfp* after acute exposure to high temperature. *p<0.05, **p<0.01, ***p<0.001, ****p<0.0001. Error bars indicate mean ± SD. Unpaired t-test.

The online version of this article includes the following source data and figure supplement(s) for figure 4:

**Source data 1.** Numerical values of DAF-16-positive nuclei plotted in panels A and B.
**Source data 2.** Numerical values of % survival plotted in panels C, D, E and F.
**Figure supplement 1.** Translocation of DAF-16 to the nucleus is impaired in *dcap-1* mutants.
**Figure supplement 2.** Translocation of DAF-16 to the nucleus is facilitated in neuronal *dcap-1* overexpressing worms.

precursors) are detrimental during development, but have an opposing lifespan-extending effect during adulthood.

## Neuronal DCP1 affects wing imaginal disc and adult wing development

While studying elav:GAL4 >UAS:IRDCP1 flies we noticed that neuronal DCP1 knockdown causes a characteristic phenotype, defined by malformed/unexpanded wings (*Figure 5E*) in ~50% of the population at 25˚C (n = 178 for GD31442 and n = 244 for KK105638). This effect did not correlate with the efficiency of the knockdown, since all flies exhibited the same levels of DCP1 mRNA regardless of their wing phenotype (*Figure 5—figure supplement 1*), neither with their lifespan, which was short in all cases. Nonetheless, penetrance reached a remarkable 95% at 29˚C (n = 167). Notably, wing malformations always coincided with the presence of additional morphological aberrations, which include an unexpanded thorax (judged by the crossing of postscutellar bristles) and the lack of cuticle tanning (melanisation and hardening) (*Figure 5E and F*). Such defects indicate that mRNA decapping in the nervous system can affect development in non-neuronal cells of *D. melanogaster*. Like most external structures of the mature fly, adult wings derive from imaginal discs, which are created during larval development and undergo extreme changes during metamorphosis (*Fristrom, 1993*). To examine whether the origins of wing malformation can be traced in early events of

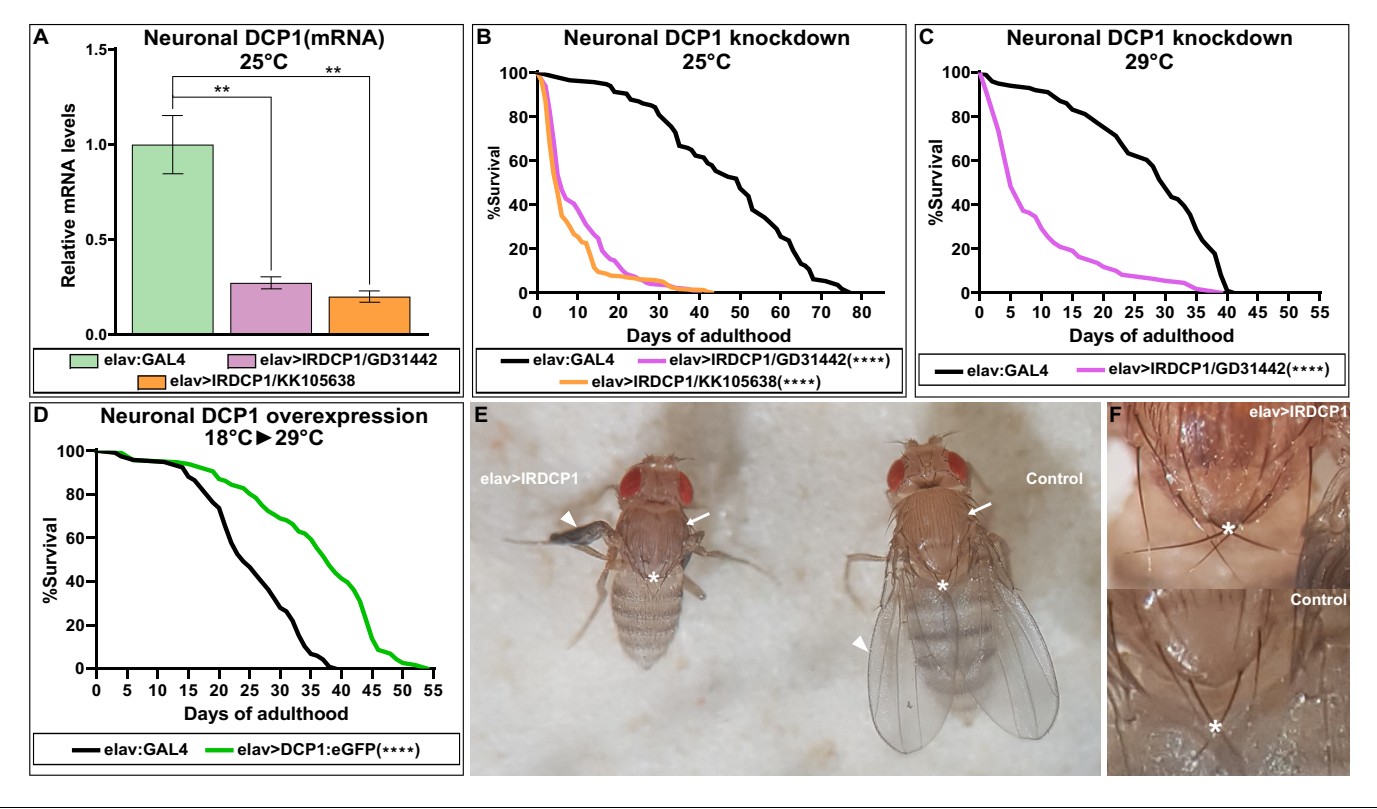

**Figure 5.** Neuronal DCP1 affects lifespan and adult wing expansion in *D. melanogaster*. (A) DCP1 mRNA levels in heads of two fly strains subjected to neuronal DCP1 knockdown. (B, C) Lifespan of flies subjected to neuronal DCP1 knockdown. (D) Lifespan of flies that neuronally overexpress a DCP1: eGFP fusion post-developmentally. (E, F) Malformations of adult flies subjected to neuron-specific DCP1 knockdown during their development (see text). Arrowheads point to wing morphology, arrows to the difference in cuticle tanning and asterisks to the crossing of postscutellar bristles. **p<0.01, ****p<0.0001. Error bars indicate mean ± SD. Unpaired t-test (A); Log-rank (Mantel-Cox) test (B–D).

The online version of this article includes the following source data and figure supplement(s) for figure 5:

**Source data 1.** Lifespan replicates of flies with neuronal DCP1 knockdown or overexpression.
**Source data 2.** Numeric values of relative mRNA levels plotted in panel 5A and figure supplement 2.
**Figure supplement 1.** The penetrance of unexpanded wings phenotype does not correlate with the efficiency of DCP1 knockdown.

their morphogenesis, we dissected wing imaginal discs from third instar larvae with neuronal DCP1 knockdown (elav:GAL4 >UAS:IRDCP1) and examined cell and tissue morphology by visualizing F-actin. Despite the neuronal DCP1 knockdown, 15 out of 23 isolated discs exhibited widespread structural anomalies, associated with abnormal epithelial folding, which were absent in discs from age-matched larvae of parental UAS:IRDCP1 and elav:GAL4 strains (*Figure 6A*).

To rule out any leakage in the specificity of our system that could lead to DCP1 knockdown in wing disc epithelial cells and affect their morphology cell autonomously (*Casas-Tintó et al., 2017*), we monitored elav:GAL4-driven GFP fluorescence in tissues derived from elav:GAL4 >UAS:DCP1: eGFP third instar larvae. As expected we observed strong fluorescence in the majority of neuronal brain cells but we detected no fluorescent signal in wing imaginal discs (*Figure 6B*). Additionally, we used an en:GAL4 driver to examine the effect of DCP1 deficiency in wing disc epithelial cells by knocking down its expression only in the posterior part of the disc. This did not result in any structural differences between the anterior and the posterior part of en:GAL4 >UAS:IRDCP1 wing discs, both of which had normal morphology in 12 out of 13 examined wing imaginal discs (*Figure 6C*). These results implicate neuronal DCP1 in a cell non-autonomous mechanism that uses neuronally derived signals to orchestrate wing disc development.

Interestingly, abnormal wing discs and malformed adult wings appeared with a similar frequency of 50–60% in elav:GAL4 >UAS:IRDCP1 flies grown at 25°C, suggesting a cause-and-effect relationship between them. To uncouple these two phenotypes, we performed temperature-shift

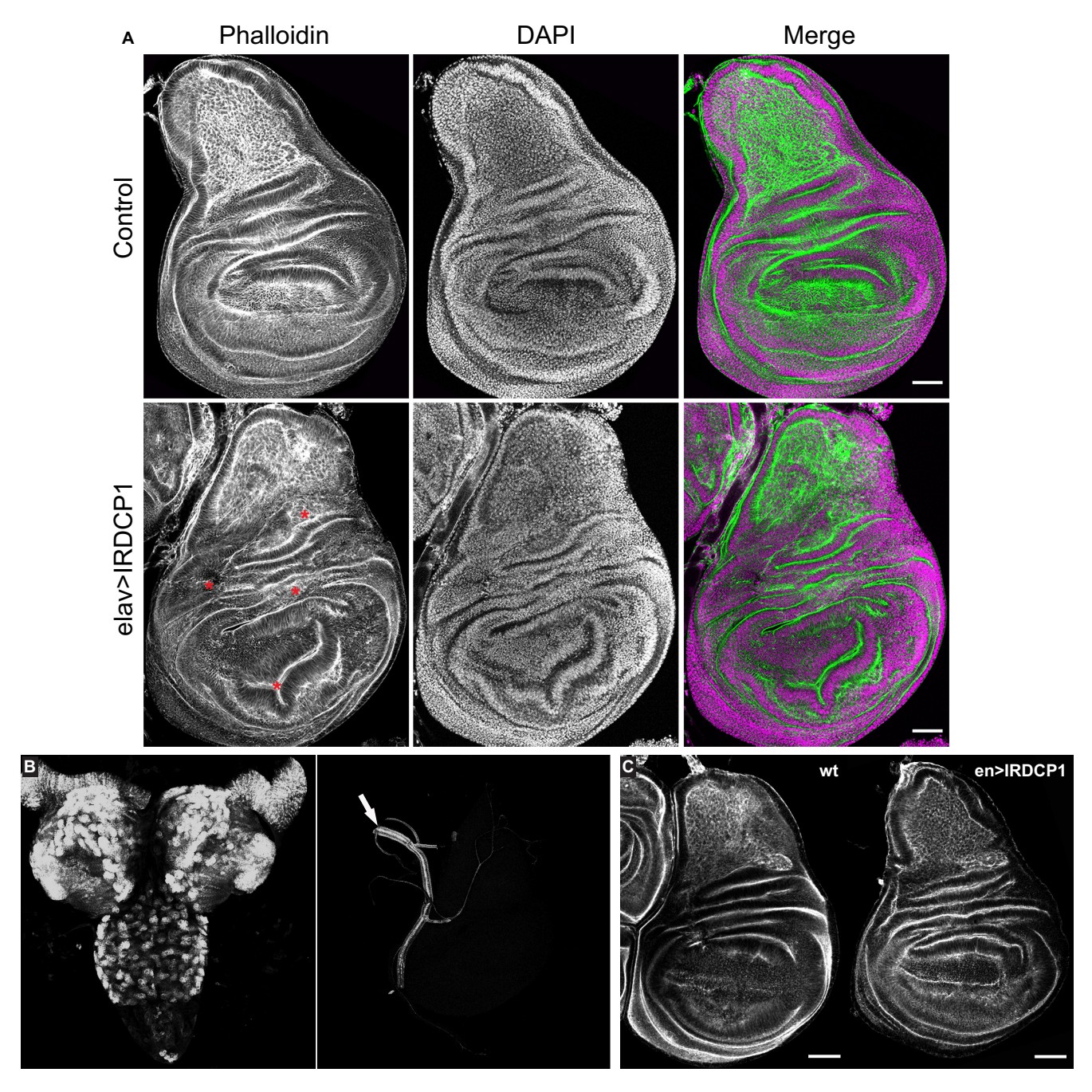

**Figure 6.** Neuronal DCP1 affects wing disk morphology. (A) Phalloidin staining of wing imaginal discs derived from animals subjected to neuron-specific DCP1 knockdown. Asterisks indicate abnormalities in epithelial folding. Green: phalloidin, Magenta: DAPI. (B) Fluorescent images of brain (left) and imaginal disc (right) dissected from elav >DCP1:eGFP larvae. Arrow shows autofluorescence in the trachea. (C) Phalloidin staining of wing discs derived from flies subjected to DCP1 knockdown only in the posterior part of wing disc epithelium (using engrailed expression driver). Anterior is to the left and posterior to the right. Scale bar = 50 µm.

experiments to temporally control the induction of neuronal DCP1 knockdown. Unsurprisingly, elav: GAL4 >UAS:IRDCP1;tubGAL80$^{ts}$ animals were superficially wild type at 18°C ('DCP1-ON' state), and exhibited both abnormal wing discs and malformed wings when grown at 29°C ('DCP1-OFF' state). Switching between DCP1-ON and OFF states at different developmental stages (*Figure 7*) revealed that restricting neuronal knockdown in embryonic and larval stages leads to the formation of

**Figure 7.** Schematic representation of thermoinducible DCP1 knock down at different developmental stages. The crucial period where DCP1 activity affects adult wing expansion is traced between the 1-day and the 3-day pupa stages.
The online version of this article includes the following figure supplement(s) for figure 7:

**Figure supplement 1.** Abnormalities in wing discs of larvae subjected to neuron-specific DCP1 knockdown at 29°C.

Schematic data (active DCP1 state per developmental stage; pink = DCP1-OFF at 29°C, green = DCP1-ON at 18°C):

| Active state per stage | Embryos | Larvae | 1-day Pupae | 2-day Pupae | 3-day Pupae | 5-day Pupae | Adults | Expanded | Unexpanded |
|---|---|---|---|---|---|---|---|---|---|
| OFF→ON (first) | OFF | OFF | OFF | OFF | OFF | OFF | OFF | 8 (5%) | 159 (95%) |
|  | OFF | OFF | OFF | OFF | OFF | OFF | ON | 0 (0%) | 23 (100%) |
|  | OFF | OFF | OFF | OFF | OFF | ON | ON | 32 (91%) | 3 (9%) |
|  | OFF | OFF | OFF | OFF | ON | ON | ON | 35 (100%) | 0 (0%) |
|  | OFF | OFF | OFF | ON | ON | ON | ON | 68 (100%) | 0 (0%) |
|  | OFF | OFF | ON | ON | ON | ON | ON | 87 (100%) | 0 (0%) |
|  | OFF | OFF | ON | ON | ON | OFF | OFF | 47 (100%) | 0 (0%) |
| ON→OFF (first) | ON | ON | ON | ON | ON | ON | ON | 58 (100%) | 0 (0%) |
|  | ON | ON | ON | ON | ON | OFF | OFF | 46 (100%) | 0 (0%) |
|  | ON | ON | ON | ON | OFF | OFF | OFF | 36 (100%) | 0 (0%) |
|  | ON | ON | ON | OFF | OFF | OFF | OFF | 48 (51%) | 46 (49%) |

abnormal wing discs (five out of nine examined wing imaginal discs displayed abnormalities in tissue folding and F-actin accumulation) (*Figure 7—figure supplement 1*) but does not affect the phenotype of adult wings. Congruently, starting knockdown from the white pupa stage, when wing discs are already formed is sufficient for causing malformed wings. Therefore, abnormalities in wing imaginal discs tissue pattern and aberrant adult wing morphology represent two unconnected outcomes of neuronal DCP1 deficiency, which occur independently at different developmental stages. Of note, we were not able to induce malformation of adult wings by switching to DCP1-OFF state past the 2-day pupa stage, while switching to DCP1-ON after the 3-day pupa stage resulted in adults with malformed wings (*Figure 7*). Taking into account, the potential latency between any temperature shift and the subsequent effect in gene expression, we can safely place the crucial period where neuronal DCP1 activity impacts adult wing development between the white pupa and the 3-day pupa stages. Supporting this notion, switching to DCP1-ON at white pupa and back to DCP1-OFF at pharate pupa stage results in adults with normal wings (*Figure 7*).

# Discussion

Decapping is an integral step of the 5'−3' mRNA decay pathway in eukaryotic cells that is vital for quality control and regulation of cellular gene expression, with effects on several physiological traits. In this study, we have shown that neuronal activity of DCAP-1/DCP1, which is the regulatory subunit of the core mRNA decapping enzyme, participates in the inter-tissue crosstalk that regulates ageing through neurosecretion. Using appropriate strains designed to express DCAP-1::GFP fusion protein in a tissue-selective manner, we found that neuronal overexpression is sufficient to extend nematode lifespan, in a DCAP-2-dependent manner, regardless of DCAP-1 availability in other tissues. A similar but far weaker effect is evident as a result of intestinal expression, whereas muscle- or hypodermal-specific *dcap-1::gfp* transgenes had no impact on longevity. These data suggest that DCAP-1 acts predominantly in neurons to influence lifespan, and infer the involvement of neuroendocrine signaling in the underlying mechanism. Several studies in *C. elegans* have revealed the cell non-autonomous role of neuroendocrine signaling pathways in the regulation of lifespan, stress resistance, innate immunity and protein homeostasis, with IIS playing a prominent role among them (*van Oosten-Hawle and Morimoto, 2014*). Indeed, we found that the effect of neuronal *dcap-1* overexpression on lifespan is exerted through the action of IIS and the downstream transcription factor DAF-16/FOXO. More specifically, DCAP-1 activity in the nervous system controls the expression levels of *ins-7* gene, encoding an insulin-like peptide that is upregulated during ageing and acts as a DAF-2/InsR agonist.

Loss of *ins-7*, through gene mutations or RNAi, can extend nematode lifespan by lowering IIS and activating DAF-16, whereas intestinally expressed *ins-7* shortens lifespan (*Murphy et al., 2003*; *Murphy et al., 2007*). We found that short-lived *dcap-1* mutant adults show higher mRNA levels of *ins-7*, compared to wt animals, whereas *dcap-1* overexpression in neurons, significantly hampers the age-dependent mRNA upsurge of *ins-7*. Elevated *ins-7* mRNA levels in *dcap-1* mutants coincide with increased transcription of *ins-7* in their intestine, despite the fact that *ins-7* is predominantly expressed in neurons (*Pierce et al., 2001*; *Murphy et al., 2007*). Overall, our data strongly implied the involvement of a neuroendocrine mechanism leading to the hypothesis that overexpression of neuronal *dcap-1* extends lifespan through IIS intertissue signaling initiated by lower levels of neuronal INS-7. This hypothesis is supported by the following evidence: a) We demonstrated post-transcriptional stabilization of neuronal *ins-7* mRNA in decapping mutants, b) deletion of *ins-7* gene suppresses the induction of *ins-7p::gfp* in the intestine of *dcap-1* mutants, in a similar way to *unc-31/* CAPS mutants, where neuropeptide secretion is strongly inhibited, c) *ins-7* RNAi in a neuronal RNAi-sensitive background increases lifespan of wt animals but fails to further extend the already long life of worms overexpressing neuronal *dcap-1*.

Lower neurosecretion of this DAF-2 agonist in *dcap-1* overexpressing animals could subsequently reduce IIS and activate DAF-16 in intestinal cells, ultimately downregulating the expression of intestinal *ins-7*. This feedback regulatory loop has been proposed to guide FOXO-to-FOXO signaling and orchestrate organismal ageing, through the action of DAF-16 in other tissues. Since *ins-7* gene is subjected to negative regulation by DAF-16 in the intestine, but not neurons (*Murphy et al., 2007*), we argued that transcriptional induction of intestinal *ins-7* in *dcap-1* mutants could result from alterations of DAF-16 activity. Indeed, we monitored that translocation of DAF-16 to the nucleus, is delayed in all tissues of *dcap-1* worms, during normal conditions and the early stages of heat-shock, while overexpression of neuronal *dcap-1* enhanced the localization of DAF-16 in both cases. Oddly, the longevity effect of neuronal *dcap-1* overexpression correlated with increased stress resistance in mid-aged animals, but not in young adults. This difference probably reflects the greater need of ageing worms for protective mechanisms toward accumulating stress factors (*Walther et al., 2015*). DAF-16 was recently reported to translocate to the nucleus of aged animals in order to stabilize their transcriptome, as a response to age-associated cellular stress (*Li et al., 2019*). Alternatively, excess neuronal DCAP-1 activity could maintain proper protein translation and homeostasis in aged neurons, preserving nervous system integrity during ageing. A thought-provoking study reported that deficiency of decapping enhancer EDC-3 extends lifespan in *C. elegans* by suppressing protein synthesis in neurons, due to sequestration of translation factors into P-bodies, which then triggers stress response through multiple longevity effectors (*Rieckher et al., 2018*). Although counterintuitive, this contrasting effect of DCAP-1 and EDC-3 on lifespan is in line with the role of decapping activators that function to provide specificity toward diverse mRNA substrates through particular interactions

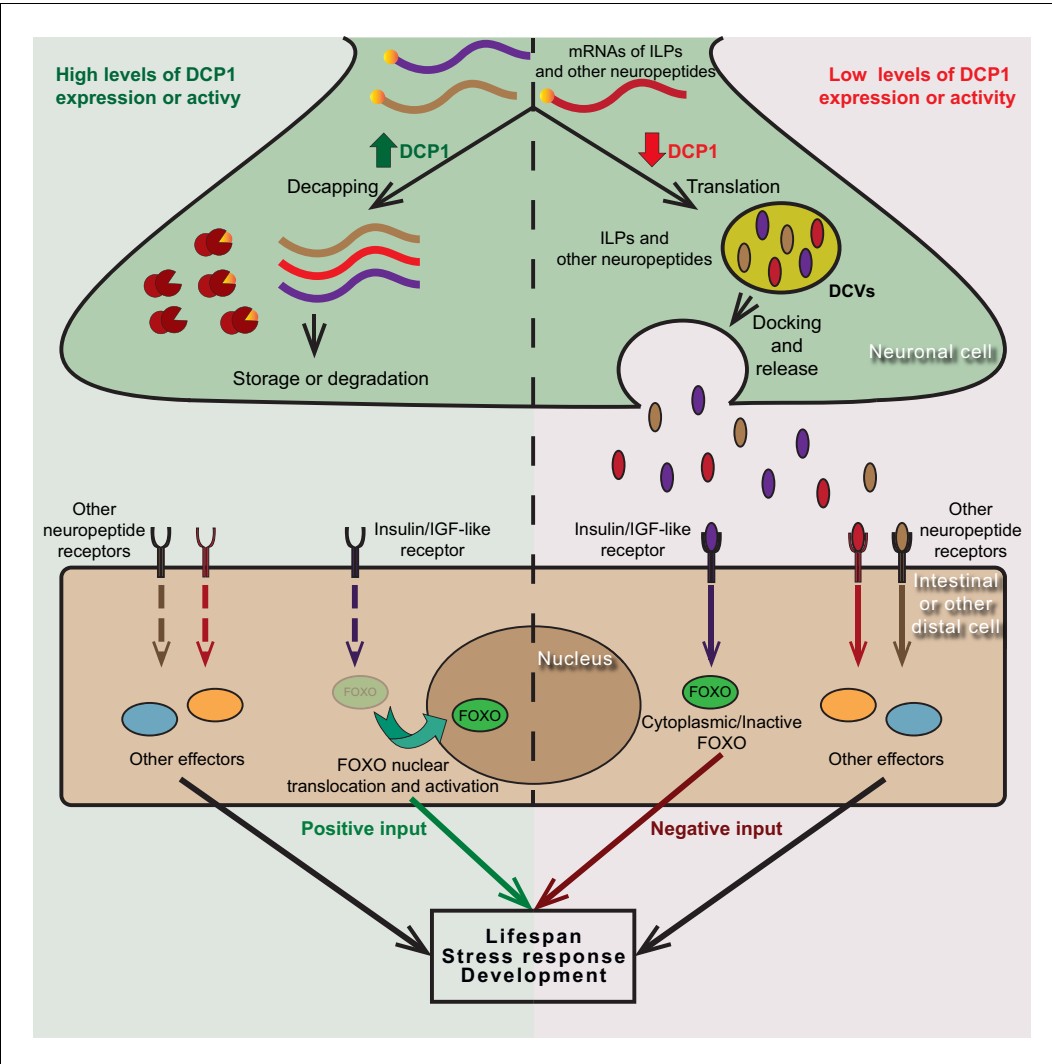

**Figure 8.** Putative model of DCP1-mediated regulation of neuroendocrine signalling that controls lifespan, stress response and development. DCP1 activity or availability in neuronal cells controls the equilibrium between translation and storage/degradation of mRNAs encoding for insulin-like peptides (ILPs) or other neuropeptides. High levels of DCP1 promote decapping, storage and degradation, while low levels favour mRNA translation and secretion the resulting ILPs or neuropeptides through DCVs. Secreted ILPs/neuropeptides bind to Insulin/IGF-like receptor or their corresponding receptor in distal cells and control the activity of downstream effectors (like FOXO transcription factor), ultimately affecting lifespan, stress response and developmental events.

with them or with other decapping enhancers (*He et al., 2018*). Moreover, the mechanism by which neuronal overexpression of *dcap-1* extends lifespan is completely dependent on DCAP-2 activity, and cannot be attributed to secondary effects emerging from the increased P-body formation that is evident in both wt and *dcap-2* backgrounds (*Supplementary file 1*).

The fact that regulation of ageing via neurons appears to be conserved in *D. melanogaster* (*Tatar et al., 2001*; *Libert et al., 2007*), combined with the strong evolutionary conservation of the 5′−3′ mRNA decay prompted us to investigate the connection between neuronal decapping and ageing in the fruit fly. We showed that neuron-specific knockdown of the fly *dcap-1* orthologue DCP1 greatly reduces lifespan, while its neuronal overexpression leads to a significant induction of longevity. The latter though applies only when overexpression occurs post-developmentally. This suggests that high levels of neuronal DCP1 are harmful during development but beneficial for lifespan during adult life. The fly genome encodes eight insulin-like peptides (DILP1-8), four of which are expressed in neurons and are linked with lifespan regulation (*Kannan and Fridell, 2013*).

Moreover, the exoribonuclease Pacman/XRN1, which acts downstream of decapping in the 5′–3′ mRNA decay pathway, has been reported to regulate the levels of secreted molecules in *D. melanogaster*, including insulin-like peptide DIPL8 and neuropeptide NPLP2 (*Jones et al., 2016*). This indicates that mRNA decapping may play an active and extended role in the control of neurosecretion in the fly. This view is in agreement with our finding that neuron-specific knockdown of DCP1 causes structural abnormalities in the epithelium of larval wing imaginal discs and impairs the execution of the post eclosion program that leads to wing expansion in adults.

Of note, all morphological traits of neuronal DCP1-deficient flies are connected with processes that take place in newly eclosed adults, during the late phases of the last ecdysis. At this stage, release of neuropeptide bursicon (also known as the tanning hormone) from neurons of the subesophageal and abdominal ganglia orchestrates the execution of a posteclosion program, characterized by a combination of behavioral and anatomical changes that act in concert and ultimately lead to wing expansion and cuticle tanning (*Peabody et al., 2008*; *White and Ewer, 2014*). A recent study revealed that rickets, the bursicon receptor, functions in a set of insulin-like peptide ILP7-secreting peptidergic neurons of the ventral nervous system to regulate the tanning of *Drosophila* adult cuticle (*Flaven-Pouchon et al., 2020*). Mutants defective for bursicon or rickets, fail to induce both behavioral and anatomical aspects of this program and consequently form adults with unexpanded wings, unexpanded thorax and untanned cuticle (*Dewey et al., 2004*; *Baker and Truman, 2002*). The striking similarity to these mutants strongly suggests that flies subjected to neuronal DCP1 knockdown exhibit some kind of impairment in bursicon signaling. In favor of this notion, our initial measurements show a trend for lower bursicon and rickets mRNA levels in heads dissected from newly eclosed adults when neuronal DCP1 is knocked-down (*Supplementary file 2*). However, our data cannot support a direct connection between DCP1 levels and bursicon activity during the last ecdysis, since DCP1 knockdown at that point does not affect wing expansion. It is rather more likely that neuronal DCP1 activity during earlier stages of pupal development is essential for the function of an intermediate factor, which subsequently affects bursicon signaling and hence the corresponding posteclosion program. A similar mechanism could affect the morphology of wing imaginal discs during neuronal DCP1 knockdown, as bursicon expression is also detected during early larval development (*Honegger et al., 2008*).

Collectively, our data have uncovered an evolutionarily conserved role for the regulatory subunit of the mRNA decapping enzyme, DCAP-1/DCP1, in the modulation of neuroendocrine signaling mechanisms that govern developmental processes and ageing (*Figure 8*). This is in line with previous work that has implicated the decapping enzyme and its regulators in the control of local translation at the synapse, synaptic plasticity and related modifications of neuronal activity in *Drosophila* and mammals (*Barbee et al., 2006*; *Hillebrand et al., 2010*; *Zeitelhofer et al., 2008*). Taken together, such findings reestablish our understanding concerning mRNA decapping factors which should no longer be considered as mere housekeeping genes that serve in the indiscriminate degradation of mRNA molecules, but as actual gene expression regulators that have active roles in the control of neuronal function.

## Materials and methods

**Key resources table**

| Reagent type (species) or resource | Designation | Source or reference | Identifiers | Additional information |
|---|---|---|---|---|
| Gene (*Caenorhabditis elegans*) | *dcap-1* | www.wormbase.org | CELE_Y55F3AM.12 | WormBase ID: WBGene00021929 |
| Gene (*Caenorhabditis elegans*) | *dcap-2* | www.wormbase.org | CELE_F52G2.1 | WormBase ID: WBGene00003582 |
| Gene (*Caenorhabditis elegans*) | *daf-16* | www.wormbase.org | CELE_R13H8.1 | WormBase ID: WBGene00000912 |

*Continued on next page*

*Continued*

| Reagent type (species) or resource | Designation | Source or reference | Identifiers | Additional information |
|---|---|---|---|---|
| Gene (*Caenorhabditis elegans*) | *ins-7* | www.wormbase.org | CELE_ZK1251.2 | WormBase ID: WBGene00002090 |
| Gene (*Drosophila melanogaster*) | DCP1 | www.flybase.org | CG11183 | FlyBase ID: FBgn0034921 |
| Strain, strain background (*Escherichia coli*) | OP50 | *Caenorhabditis* Genetics Center (CGC) | OP50 | For standard NGM plates |
| Strain, strain background (*Escherichia coli*) | HT115 | *Caenorhabditis* Genetics Center (CGC) | HT115(DE3) | For RNAi plates |
| Strain, strain background (*Caenorhabditis elegans*) | *C. elegans* strains used in this study | This study | | *Supplementary file 3* |
| Strain, strain background (*Drosophila melanogaster*) | *D. melanogaster* strains used in this study | This study | | *Supplementary file 4* |
| Recombinant DNA reagent | Primers used in this study | This study | | *Supplementary file 5* |
| Recombinant DNA reagent | promoterless *dcap-1::gfp* | *Borbolis et al., 2017* DOI: 10.1098/rsob.160313 | PS#302 | pBluescript KS(+); 4181 bp *dcap-1::gfp* |
| Recombinant DNA reagent | *unc-119p::dcap-1::gfp* | *Borbolis et al., 2017* DOI: 10.1098/rsob.160313 | PS#293 | pBluescript KS(+); 2200 bp *unc-119p*; 4181 bp *dcap-1::gfp* |
| Recombinant DNA reagent | *rab-3p::dcap-1::gfp* | This study | PS#345 | pBluescript KS(+); 1200 bp *rab-3p*;4181 bp *dcap-1::gfp* |
| Recombinant DNA reagent | *ges-1p::dcap-1::gfp* | *Borbolis et al., 2017* DOI: 10.1098/rsob.160313 | PS#328 | pBluescript KS(+); 1541 bp *ges-1p*; 4181 bp *dcap-1::gfp* |
| Recombinant DNA reagent | *hlh-1p::dcap-1::gfp* | *Borbolis et al., 2017* DOI: 10.1098/rsob.160313 | PS#326 | pBluescript KS(+);3100 bp *hlh-1p*; 4181 bp *dcap-1::gfp* |
| Recombinant DNA reagent | *col-10p::dcap-1::gfp* | *Borbolis et al., 2017* DOI: 10.1098/rsob.160313 | PS#364 | pBluescript KS(+);2000 bp col-10p; *4181 bp dcap-1::gfp* |
| Recombinant DNA reagent | *ges-1p::gfp* | This study | PS#176 | pPD95.77;1541 bp *ges-1p* |
| Recombinant DNA reagent | *unc-119p::ins-7* | This study | PS#478 | pBluescript SK II; 2200 bp *unc-119p*; 1082 bp *ins-7* |
| Recombinant DNA reagent | *UAS:DCP1:eGFP* | This study | PS#394 | pUASTattB; 1116 bp *dcp1*; 924 bp *egfp* |
| Recombinant DNA reagent | *daf-16 RNAi* | This study | PS#48 | pPD129.36(L4440);1721 bp genomic DNA |
| Recombinant DNA reagent | *daf-2 RNAi* | This study | PS#36 | pPD129.36(L4440);1393 bp genomic DNA |
| Recombinant DNA reagent | *ins-7 RNAi* | This study | PS#452 | pPD129.36(L4440);743 bp genomic DNA |
| Commercial assay or kit | QIAprep spin Miniprep Kit (50) | QIAGEN | QIA.27104 | |
| Commercial assay or kit | Qiaquick PCR purification kit | QIAGEN | QIA.28104 | |
| Commercial assay or kit | Nucleospin RNA XS | Macherey-Nagel | 740902.50 | |
| Commercial assay or kit | FIREScipt RT cDNA Synthesis kit | SOLIS BIODYNE | 06-15-00200 | |

*Continued on next page*

*Continued*

| Reagent type (species) or resource | Designation | Source or reference | Identifiers | Additional information |
|---|---|---|---|---|
| Commercial assay or kit | Maxima H Minus First Strand cDNA Synthesis Kit | ThermoFisher Scientific | K1682 | |
| Commercial assay or kit | KAPA SYBR FAST 2X MasterMix, Universal | Kapa Biosystems | KK4602 | |
| Commercial assay or kit | DNA oligos synthesis | IDT | | |
| Chemical compound | Tri Reagent | Sigma-Aldrich | T9424 | |
| Chemical compound | FUdR | Sigma-Aldrich | F0503 | |
| Software, algorithm | Prism 8 | Graphpad Software | N/A | www.graphpad.com/scientific-software/prism/ |
| Software, algorithm | Fiji | Image J | N/A | imagej.net/Fiji |
| Software, algorithm | Illustrator CS6 | Adobe | | www.adobe.com/products/illustrator.html |
| Software, algorithm | Photoshop CS6 | Adobe | | www.adobe.com/products/photoshop.html |
| Software, algorithm | HCI Imaging | HAMAMATSU | | hcimage.com/ |
| Software, algorithm | Leica Application Suite Advanced Fluorescence | Leica | | |

## *C. elegans* strains and culture conditions

Standard methods of culturing and handling worms were used (*Brenner, 1974*). Worms were cultured on NGM plates seeded with *Escherichia coli* OP50. RNAi experiments were performed on plates seeded with HT115(DE3) bacteria, expressing dsRNA for the indicated gene. For their preparation, bacteria transformed with the corresponding RNAi constructs were grown overnight at 37°C in LB medium with ampicillin (50 µg/ml) and tetracycline (10 µg/ml). On the following day, fresh cultures with ampicillin were induced with 1 mM isopropyl β-D-1-thiogalactopyranoside (IPTG) and seeded on RNAi plates. Bacteria carrying the empty vector (pL4440) were treated likewise and used as control cultures. Whenever it was deemed necessary, bacterial lawn was killed by UV irradiation and plates where supplemented with 50 µg/ml 5-fluoro-2'-deoxyuridine (FUdR) to prevent progeny growth. Wild type Bristol N2 and mutant strains were provided by the Caenorhabditis Genetics Center (CGC, University of Minnesota), which is supported by NIH Office of Research Infrastructure Programs (P40 OD010440). *dcap-1(tm3163)* mutant strain was provided by the Mitani Laboratory through the National Bio-Resource of the MEXT, Japan. All *C. elegans* strains used in this study are presented in *Supplementary file 3*. All single mutants were crossed at least three times with N2 and double mutants were generated by crossing the corresponding strains. Relevant mutations were tracked in $F_2$ progeny either by PCR or phenotypic selection. Primers used for genotyping are presented in *Supplementary file 5*. BRF410 transgenic animals were generated by microinjection of plasmid DNA into to the gonad of *dcap-1(tm3163)* young adults, using *rol-6(su1006)* as a co-transformation marker. Multiple lines were observed and screened for the representative expression pattern (*Mello and Fire, 1995*). To enhance the effectiveness of neuronal RNAi by feeding, strains that express SID-1 dsRNA transporter ectopically in neurons were used (*Calixto et al., 2010*).

## *D. melanogaster* strains and culture conditions

All fly stocks and crosses were raised on standard cornmeal medium and kept at 25°C except for temperature shift experiments. elav:GAL4 (P{w[+mW.hs]=GawB}elav[C155] – BL#458) and en:GAL4 (y[1] w[*]; P{w[+mW.hs]=en2.4 GAL4}e16E – BL#30564) driver strains were obtained by Bloomington *Drosophila* Stock Center (BDSC). UAS:IRDCP1 strains were obtained from Vienna *Drosophila* Resource Center (VDRC – ID GD31442 and KK105638). UAS:DCP1:eGFP was generated by microinjection of the appropriate construct in embryos that carry the *attP* element in

the 86Fb site (BestGene). All *D. melanogaster* strains used in this study are presented in *Supplementary file 4*.

## Constructs

For *C. elegans* RNAi constructs, a corresponding gene-specific genomic DNA fragment was inserted in L4440 feeding vector (pPD129.36) (*Timmons and Fire, 1998*). In the case of *daf-16* and *daf-2*, a 1.7 kb HindIII/XhoI and a 1.4 kb BamHI/XhoI fragment were used, respectively. For *ins-7* RNAi, a 750 bp fragment was amplified by PCR and inserted as an XbaI/XhoI fragment. To generate the *rab-3p:: dcap-1::gfp* construct, a 1.2 kb DNA fragment was amplified from genomic DNA by PCR and inserted as a PstI/BamHI fragment in a previously described promoterless *dcap-1::gfp* vector (*Borbolis et al., 2017*). For the *unc-119p::ins-7* construct, a 2.2 kb fragment, corresponding to the promoter of *unc-119*, was amplified from genomic DNA by PCR and cloned into pBluescript SK II as an XhoI/PstI fragment, followed by a 1.1 kb fragment containing *ins-7* coding region and its 3' UTR, which was amplified from genomic DNA by PCR and cloned as PstI/XbaI fragment. For the *ges-1p:: gfp* construct, a 1.5 kb PstI fragment, amplified from genomic DNA by PCR was cloned into pPD95.77 *gfp*-containing vector. *D. melanogaster*'s DCP1:eGFP construct was generated by inserting a 1.1 kb EcoRI/BglII fragment containing DCP1 coding sequence (amplified from cDNA clone GH04763 *Drosophila* Genomics Resource Center) and a 924 bp BglII/XhoI fragment containing the eGFP sequence (amplified from vector pEGFP-N1 (Clonetech)) into pUASTattB vector *Drosophila* Genomics Resource Center). All primers used for the amplification of insert fragments are shown in *Supplementary file 5*.

## Lifespan assays

*C. elegans* lifespan analysis was conducted at 20°C or at 25°C as described previously (*Syntichaki et al., 2007*). Briefly, mid to late L4 larvae of each strain were transferred to NGM plates (30–40 per plate) seeded with OP50 or HT115 (for RNAi) bacteria and moved to fresh plates every 2–4 days. Viability was scored daily and worms that failed to respond to stimulation by touch were considered dead. Bagged or raptured worms and animals that crawled off the plates are referred as censored in the analysis. In *D. melanogaster* lifespan assays, the desired male flies were crossed with appropriate virgin females and F1 progeny were divided in vials of 20–30 individuals. Vials were incubated at the desired temperature inclined in order to prevent drowning. Flies were transferred to fresh ones every 2–4 days and viability was scored daily. Flies that died accidentally or escaped were scored as censored. All crosses and lifespan assays were carried out at 29°C, with the exception of post-developmental DCP1 overexpression where crosses were performed at 18°C and progeny were transferred at 29°C as 1 day old adults. Statistical analysis was performed by comparing each population to the appropriate control and p-values were determined using the log-rank (Mantel-Cox) test.

## RNA isolation and quantitative reverse transcription PCR (qRT-PCR)

Total RNA was prepared form frozen worm pellets (200–300 worms per sample) or fly heads (at least 10 per sample) of the indicated genetic background and age, using Tri Reagent (Sigma-Aldrich) or NucleoSpin RNA XS kit (Macherey-Nagel). *C. elegans* worms were cultured at 20°C in the presence of 50 µM FUdR to avoid progeny growth. At least three populations were harvested and analyzed independently in each experiment. *D. melanogaster* flies were raised in the indicated temperature until they reached the desired age or developmental stage where their heads were dissected and frozen. Quality and quantity of RNA samples were determined using Nanodrop 2000 Spectrophotometer (Thermo Scientific). Reverse transcription was carried out with FIREScipt RT cDNA Synthesis KIT (Solis BioDyne) and quantitative PCR was performed using KAPA SYBR FAST Universal Kit (Kapa Biosystems) in the MJ MiniOpticon system (BioRad). Relative amounts of mRNA were determined using the comparative Ct method for quantification and each sample was independently normalized to its endogenous reference (*ama-1* for *C. elegans*, Rpp20 for *D. melanogaster*). Gene expression data are presented as the mean fold change ± SEM of all biological replicates relative to control. Statistical analysis was performed by comparing each sample to the appropriate control and p-values were determined using unpaired *t*-test. Primer sequences used for qRT-PCR are shown in *Supplementary file 5*.

## Microscopy

Microscopic analysis of fluorescent *C. elegans* worms was performed by monitoring levamisole-treated animals mounted on 3% agarose pads on glass microscope slides. Images were captured by confocal microscopy using a Leica TCS SP5 II laser scanning confocal imaging system on a DM6000 CFS upright microscope and a 10x objective or by fluorescent microscopy using a Leica DMRA upright fluorescent microscope equipped with a Hamamatsu ORCA-flash 4.0 camera and 20x or 40x objectives. Microscopy settings were kept stable throughout each experiment. All worms were cultured at 20°C in the presence of 50 μM FUdR to avoid progeny growth. DAF-16::GFP and DAF-16::RFP animals were fixed with 4% PFA in 1xPBS for 10 min and washed twice with M9 buffer prior to visualization to avoid DAF-16 translocation as result of handling. *ins-7p::gfp* and *ges-1p::gfp* driven fluorescence intensity was measured with ImageJ 1.52p (Fiji). For *ins-7p::gfp* sum slices projections of z-stacks were used. Fluorescence of 20–30 worms was measured for each strain and time point. DAF-16::GFP positive nuclei were counted manually using maximum intensity projections of z-stacks generated in ImageJ. DAF-16::RFP-positive nuclei were counted using Fiji macro scripts, which we developed for this purpose. Approximately 20 worms per strain and time point were scored. In all cases, the mean ± SEM of calculated values in each time point was plotted. Statistical analysis was performed by comparing each sample to the appropriate control in the same time point and p-values were determined using unpaired *t*-test. Microscopic analysis of *D. melanogaster* imaginal discs and brains was performed after dissection in 1xPBS using fine forceps. The tissues were fixed with 4% PFA in 1xPBS for 10–20 min in RT and washed with 1xPBS. F-actin was labelled using Alexa-Fluor-488 phalloidin. Mounting of tissues was performed using Vectashield. Images were captured using Leica SP5 II laser scanning confocal imaging system on a DMI6000 CFS inverted microscope and a 20x objective. Shown images are single optical sections.

## Stress resistance assays

Heat shock assays in 1-day-old worms were performed by shifting synchronous populations from 20°C to 35°C for 6 hr. In the case of 9-day-old adults, worms were incubated at 35°C for 8 hr in plates seeded with UV-killed bacteria, supplemented with 50 μg/ml FUdR. Survival was scored after 16 hr of recovery at 20°C. Acute oxidative stress was inflicted by transferring 1-day-old worms to plates seeded with UV-killed bacteria containing 5 mM sodium arsenite. Survival was scored after 24 hr. For chronic oxidative stress 1-day-old adults were transferred to plates seeded with UV-killed bacteria containing 2.5 mM sodium arsenite and viability was scored daily. In all cases, the mean ± SEM of at least three independent experiments was plotted. Statistical analysis was performed by comparing each sample to the appropriate control and p-values were determined using unpaired *t*-test.

## Statistics

Statistical analysis in all case was performed with GraphPad Prism version 8.0.0 for Windows (GraphPad Software, San Diego, California USA, www.graphpad.com.). Significance is depicted as follows: **** $p < 0.0001$; *** $p = 0.0001$–$0.001$; ** $p = 0.001$–$0.01$; * $p = 0.01$–$0.05$; ns indicates not significant with p-value $\geq 0.05$.

## Acknowledgements

Worm strains were provided by the Caenorhabditis Genetics Center (CGC, University of Minnesota), which is supported by NIH Office of Research Infrastructure Programs (P40 OD010440) and the Mitani Laboratory through the National Bio-Resource of the MEXT, Japan. Fly stocks obtained from the Bloomington *Drosophila* Stock Center (NIH P40OD018537) were used in this study. Transgenic RNAi fly stocks were obtained from the Vienna *Drosophila* Resource Center (VDRC, www.vdrc.at). We thank the BRFAA Imaging Unit for support with confocal microscopy. We are grateful to all members of PS and CZ laboratories for their help and input.

## Additional information

### Funding

| Funder | Grant reference number | Author |
|---|---|---|
| European Research Council | 201975 | Popi Syntichaki |
| Fondation Sante | 2018 | Popi Syntichaki |

The funders had no role in study design, data collection and interpretation, or the decision to submit the work for publication.

### Author contributions

Fivos Borbolis, Data curation, Investigation, Visualization, Methodology; John Rallis, George Kanatouris, Nikolitsa Kokla, Antonis Karamalegkos, Christina Vasileiou, Katerina M Vakaloglou, Investigation; George Diallinas, Dimitrios J Stravopodis, Resources; Christos G Zervas, Conceptualization, Resources, Data curation, Supervision, Investigation; Popi Syntichaki, Conceptualization, Resources, Data curation, Supervision, Funding acquisition, Validation, Investigation, Methodology, Project administration

### Author ORCIDs

Fivos Borbolis (iD) https://orcid.org/0000-0001-6559-1356
George Diallinas (iD) http://orcid.org/0000-0002-3426-726X
Christos G Zervas (iD) https://orcid.org/0000-0003-0531-9515
Popi Syntichaki (iD) https://orcid.org/0000-0001-9536-8905

### Decision letter and Author response

Decision letter https://doi.org/10.7554/eLife.53757.sa1
Author response https://doi.org/10.7554/eLife.53757.sa2

## Additional files

### Supplementary files

• Supplementary file 1. Neuronal overexpression of a *dcap-1::gfp* fusion results in its aggregation to P-body like structures. Representative fluorescent images of wild type and *dcap-2(ok3032)* worms that express a *dcap-1::gfp* fusion under the control of the pan-neuronal *unc-119* promoter. Arrowheads point to P-body like structures. Scale bar = 20 µm.

• Supplementary file 2. mRNA levels of bursicon and rickets. Relative mRNA levels of bursicon and its receptor rickets in heads dissected form newly eclosed adults subjected to neuron-specific DCP1 knockdown at 29°C.

• Supplementary file 3. List of *C. elegans* strains used in this study.

• Supplementary file 4. List of *D. melanogaster* strains used in this study.

• Supplementary file 5. List of primers used in this study.

• Transparent reporting form

### Data availability

All data generated or analysed during this study are included in the manuscript and supporting files.

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
