## [Decision Letter]

**Acceptance summary:**

The study elegantly demonstrates that neuronal overexpression of the mRNA decapping factor DCAP-1/DCP1 is a conserved longevity factor that extends lifespan in both worms and flies. It identifies mRNAs encoding insulin like peptides as the targets of DCAP-1/DCP1 which explain the physiological and the lifespan extension effects of manipulating DCAP-1/DCP1. In summary, the study elucidates DCAP-1/DCP1 as a novel regulator of insulin like peptides to modulate longevity and metabolism.

**Decision letter after peer review:**

Thank you for submitting your article "mRNA decapping is an evolutionarily conserved modulator of neuroendocrine signaling that controls development and ageing" for consideration by *eLife*. Your article has been reviewed by three peer reviewers, and the evaluation has been overseen by a Reviewing Editor and Jessica Tyler as the Senior Editor. The following individuals involved in review of your submission have agreed to reveal their identity: Brian M Zid (Reviewer #2); Aric Rogers (Reviewer #3).

The reviewers have discussed the reviews with one another and the Reviewing Editor has drafted this decision to help you prepare a revised submission.

Summary:

The study by Borbolis and colleagues shows that neuronal overexpression of the mRNA decapping factor DCAP-1/DCP1 is a conserved lifespan factor that extends the lifespan of both *C. elegans* and *Drosophila* by inhibiting insulin signaling. It follows that the decapping machinery, generally considered to be critical for housekeeping functions, can potentially regulate differentially regulate physiological outcomes based on environmental inputs. The key conceptual advance here is the demonstration that DCP1 regulates insulin signaling (INS-7).

All three reviewers and the reviewing editor argue for the need to define the mechanism by which DCP-1 regulates insulin signaling for this study to be suitable for publication in *eLife*. Also all three reviewers have requested more data to support the hypothesis that mRNA decapping and not any other function of DCP-1 is critical for insulin signaling and longevity effects.

This could be achieved in a combination of ways including: use of reporters to measure mRNA stability, quantify mRNA levels and stability through RNA seq or other methods and genetic approaches. I outline the key concerns raised by the reviewers to substantiate this issue.

Essential revisions:

1) Reviewer 2 states: The authors propose that *dcap-1* overexpression is driving lifespan effects by decreasing neuronal INS-7 levels. The authors show that intestinal INS-7 levels are regulated transcriptionally, but that *dcap-1* does not seem to affect a neuronal INS-7 reporter transcriptionally. Is *dcap-1* affecting mRNA levels through mRNA stability? Is it changing protein levels through translatability? Is it changing INS-7 secretion? I believe differentiating between these three possibilities would significantly clarify the mechanism by which *dcap-1* affects lifespan.

2) Also reviewer 2 states: It is known that overexpressing P-body proteins such as DCP1, can lead to increased P-body formation, but even that doesn't mean there is more decapping as a number of recent articles have questioned whether mRNA decay is even taking place in P-bodies (Luo, Na and Slavoff, 2018). An alternative explanation for the lifespan extension, similar to what is postulated in Rieckher et al., 2018, is that overexpressing *dcap-1* leads to enhanced P-body formation sequestering specific translation factors into the granules, reducing INS-7 translation in these neurons, without necessitating an effect on mRNA decapping.

3) Reviewer 3 states: Decapping machinery is associated with processing bodies that are dynamic and important for adaptive responses to different conditions. Furthermore, decapping is not necessarily a death sentence for mRNA. Within P bodies, decapped mRNA may be stored and later re-capped by cytoplasmic factors. Specificity for processing specific mRNAs is made possible with different combinations of RNA binding proteins and microRNAs. Although the factors important for imparting such specificity to influence lifespan are not identified in the current study, authors do identify an important role for ILS and the ILP INS-7 for mediating longevity effects due to *dcap-*1 modulation.

4) Another issue raised is regarding the lack of mechanism by *ins-7* is regulated. Reviewer 1 states: The authors suggest that changes in promoter activity, based on the *ins-7* promoter driving GFP expression, argues in favor of DCP1 regulating *ins-7* at the level of transcription. Does this assay really distinguish transcription from post-transcriptional processes? For example, if RNA degradation pathways were downregulated with the manipulation, isn't that a post-transcriptional process that would result in the same effect on the assay used? Also, manipulations that abolish the increase in *ins-7p::gfp* fluorescence in dcp1 mutants also have greatly increased fluorescence in the control genotype (Figure 3D and G), suggesting a ceiling effect might hinder the ability to observe any changes caused by dcp1.

5) Reviewer 1 also suggests extending the genetic approaches which seem "limited without either extending the genetic studies (e.g., do manipulations of XRN1, DCP2, or other decapping components phenocopy DCP1 manipulations) or better validating the link between DCP1 and insulin signaling. Are the aging and neuroendocrine phenotypes dependent on the decapping catalytic subunit DCP2? Does deletion of *dcap-2* completely suppress the lifespan effects of *dcap-1* neuronal overexpression? This would be a way to further test whether mRNA decapping was important for these phenotypes."

---

## [Author Response]

Essential revisions:1) Reviewer 2 states: The authors propose that dcap-1 overexpression is driving lifespan effects by decreasing neuronal INS-7 levels. The authors show that intestinal INS-7 levels are regulated transcriptionally, but that dcap-1 does not seem to affect a neuronal INS-7 reporter transcriptionally. Is dcap-1 affecting mRNA levels through mRNA stability? Is it changing protein levels through translatability? Is it changing INS-7 secretion? I believe differentiating between these three possibilities would significantly clarify the mechanism by which dcap-1 affects lifespan.

We fully agree with the reviewer on these points. Based on the *ins-7::gfp* reporter, the increased *ins-7* abundance in decapping mutants seems to emanate from transcriptional induction of intestinal, and not neuronal, *ins-7*. Since this upregulation is rescued by restoring *dcap-1* in neurons and depends on neurosecretion, we suggested that loss of *dcap-1* impacts neuronal *ins-7* levels and its signaling to other tissues. Now, to address whether loss of *dcap-1* affects neuronal *ins-7* levels through mRNA stability, we have evaluated the relative mRNA levels of a neuronally expressed *ins-7* transgene *synEx478* (*unc-119p::ins-7::3’UTR_ins-7_*), in *ins-7* and *dcap-1;ins-7* mutant backgrounds (strains BRF792 and BRF793 in Supplementary file 3 of the revised manuscript). In these transgenic worms, where only neuronal INS-7 is produced, we quantified and compared the ratio of mature *ins-7* mRNA levels to its pre-mRNA levels, in order to avoid discrepancies due to chimeric expression of the multicopy extrachromosomal array. We observed a two-fold induction in both day 1 and day 9 adults of *dcap-1* compared to wt (revised Figure 3G). In contrast, the ratio of mature *eft-3* mRNA to its pre-mRNA (produced by the endogenous gene F31E3.5 and encoding the core translation elongation factor eEF1A), was similar in the two genetic backgrounds, excluding differences in pre-mRNA processing (revised Figure 3—figure supplement 7). We included these results in the revised manuscript (Results and Discussion). Collectively, our data support a stabilization of neuronal *ins-7* mRNA in decapping mutants, although we cannot yet exclude the contribution of enhanced translation or secretion of INS-7 in these mutants. We hope that the reviewer will understand the time needed to pursue definitive answers to these questions.

2) Also reviewer 2 states: It is known that overexpressing P-body proteins such as DCP1, can lead to increased P-body formation, but even that doesn't mean there is more decapping as a number of recent articles have questioned whether mRNA decay is even taking place in P-bodies (Luo, Na and Slavoff, 2018). An alternative explanation for the lifespan extension, similar to what is postulated in Rieckher et al., 2018, is that overexpressing dcap-1 leads to enhanced P-body formation sequestering specific translation factors into the granules, reducing INS-7 translation in these neurons, without necessitating an effect on mRNA decapping.

To assess the requirement of mRNA decapping activity for lifespan extension of worms overexpressing *dcap-1*, we have generated a *dcap-2* deletion mutant strain that carries the neuronal or the intestinal *dcap-1::gfp* transgene (BRF774 or BRF775 in revised Supplementary file 3 of the revised manuscript). As demonstrated in revised Figure 2E and Figure 2— source data 3, lack of DCAP-2 catalytic subunit failed to extend lifespan of these worms, signifying that mRNA decapping activity is necessary for longevity when *dcap-1* is overexpressed in these tissues. In addition, observation under microscope confirmed that these *dcap-2* transgenic worms formed P-body-like granules similarly to or even to a greater extent than wt transgenic worms (Supplementary file 1), pointing to a different mechanism from those postulated in Rieckher et al., 2018, as we mentioned in the Discussion section of the revised manuscript. We thank the reviewer for the reference of Luo et al., 2018, which was added in the Introduction of the revised manuscript.

3) Reviewer 3 states: Decapping machinery is associated with processing bodies that are dynamic and important for adaptive responses to different conditions. Furthermore, decapping is not necessarily a death sentence for mRNA. Within P bodies, decapped mRNA may be stored and later re-capped by cytoplasmic factors. Specificity for processing specific mRNAs is made possible with different combinations of RNA binding proteins and microRNAs. Although the factors important for imparting such specificity to influence lifespan are not identified in the current study, authors do identify an important role for ILS and the ILP INS-7 for mediating longevity effects due to dcap-1 modulation.

We appreciate this highly positive feedback by reviewer 3.

4) Another issue raised is regarding the lack of mechanism by ins-7 is regulated. Reviewer 1 states: The authors suggest that changes in promoter activity, based on the ins-7 promoter driving GFP expression, argues in favor of DCP1 regulating ins-7 at the level of transcription. Does this assay really distinguish transcription from post-transcriptional processes? For example, if RNA degradation pathways were downregulated with the manipulation, isn't that a post-transcriptional process that would result in the same effect on the assay used? Also, manipulations that abolish the increase in ins-7p::gfp fluorescence in dcp1 mutants also have greatly increased fluorescence in the control genotype (Figure 3D and G), suggesting a ceiling effect might hinder the ability to observe any changes caused by dcp1.

The reviewer argues that increased GFP expression driven by *ins-7* promoter in intestinal cells of *dcap-1* mutants may have been attained not by transcriptional activation but through post-transcriptional stabilization of the produced mRNA, due to downregulation of the mRNA decay pathway. Since we did not observe an analogous increase in neuronal *ins-7p::gfp* fluorescence (revised Figure 3B and Figure 3—figure supplement 1), we examined whether such a post-transcriptional process takes place only in intestinal cells of *dcap-1* worms. Thus, we generated a transgene that drives *gfp* expression under the control of the intestine-specific *ges-1* promoter and compared fluorescence in age-matched wt and *dcap-1* animals (lines BRF169 and BRF176, respectively, in revised Supplementary file 3). As shown in Figure 3—figure supplement 2 of the revised manuscript, GFP signal in *dcap-1* worms was similar to that of N2 on the first day of adulthood, contrary to *ins-7p::gfp* fluorescence which was significantly higher in *dcap-1* (revised Figure 3B and Figure 3—figure supplement 1).

Regarding the increased relative fluorescence of *ins-7::gfp* in control worms in revised Figures 3C and F (previously named Figures 3D and G), compared to Figure 3B, which reach a maximum at approximately day 6 of adulthood, we believe that it might be due to a compensatory mechanism, in order to increase *ins-7* transcription in the context of their mutant background (*unc-31* and *ins-7*, respectively). However, in days 1 and 2 of adulthood, before a plateau is reached, the relative fluorescence intensity of control and *dcap-1* strains (in revised Figures 3C and F) increases at an almost identical pace, in contrast to the 2-fold difference in intensity between control and *dcap-1* worms in Figure 3B (graph of total *ins-7p::gfp* fluorescence). In an effort to fully address the reviewer’s concern, and given that reduced *daf-16* activity can increase intestinal expression of *ins-7p::gfp* reporter (Murphy et al., 2007), we also performed *daf-*16 RNAi in *unc-31* and *dcap-1;unc-31* worms expressing *ins-7p::gfp* (wwEx66). As expected, *daf-*16 RNAi treatment further raised the threshold of GFP signal in *unc-31;wwEx66* worms, compared to empty vector RNAi, and this was independently of *dcap-1* gene function (Figure 3—figure supplement 4 in revised manuscript). These data exclude a ceiling effect that could mask the impact of *dcap-1* mutation on total fluorescence in these backgrounds. Thus, we concluded that the effect of *dcap-1* in *ins-7::gfp* fluorescence induction depends on INS-7 neurosecretion through DCVs.

5) Reviewer 1 also suggests extending the genetic approaches which seem "limited without either extending the genetic studies (e.g., do manipulations of XRN1, DCP2, or other decapping components phenocopy DCP1 manipulations) or better validating the link between DCP1 and insulin signaling. Are the aging and neuroendocrine phenotypes dependent on the decapping catalytic subunit Dcp2? Does deletion of dcap-2 completely suppress the lifespan effects of dcap-1 neuronal overexpression? This would be a way to further test whether mRNA decapping was important for these phenotypes."

We thank the reviewer for this helpful suggestion. We have published the effects of *dcap-2* deletion or RNAi-mediated knockdown of *dcap-2*, *xrn-1* and *part-1* on wt worms, which all shortened lifespan and affected developmental growth, like *dcap-1* manipulations (Rousakis et al., 2014; Borbolis et al., 2017). To further elaborate on the role of DCAP-2 on the observed longevity phenotype, we have now provided additional data using a *dcap-2* deletion strain overexpressing neuronal and intestinal DCAP-1::GFP transgenes, as deletion of *xrn-1* is lethal for worms (Newbury and Woollard, RNA, 2004). As already mentioned above in response number 2, the lifespan of this strain was indistinguishable from that of control *dcap-2* animals (revised Figure 2E and Figure 2—source data 3). We therefore established that decapping activity is required for lifespan extension of *dcap-1* overexpressing worms.